# Direct-acting antiviral resistance of Hepatitis C virus is promoted by epistasis

Hang Zhang [1], Ahmed Abdul Quadeer [1] ✉ & Matthew R. McKay [2,3] ✉

Direct-acting antiviral agents (DAAs) provide efficacious therapeutic treatments for chronic Hepatitis C virus (HCV) infection. However, emergence of drug resistance mutations (DRMs) can greatly affect treatment outcomes and impede virological cure. While multiple DRMs have been observed for all currently used DAAs, the evolutionary determinants of such mutations are not currently well understood. Here, by considering DAAs targeting the non-structural 3 (NS3) protein of HCV, we present results suggesting that epistasis plays an important role in the evolution of DRMs. Employing a sequence-based fitness landscape model whose predictions correlate highly with experimental data, we identify specific DRMs that are associated with strong epistatic interactions, and these are found to be enriched in multiple NS3-specific DAAs. Evolutionary modelling further supports that the identified DRMs involve compensatory mutational interactions that facilitate relatively easy escape from drug-induced selection pressures. Our results indicate that accounting for epistasis is important for designing future HCV NS3-targeting DAAs.

Hepatitis C virus (HCV) is the major cause of liver-associated disease and liver cancer, affecting more than 180 million people worldwide[1]. Fortunately, effective drug treatments, using direct-acting antiviral agents (DAAs), are available that achieve sustained virological response (SVR) in more than 95% of patients[2,3]. However, drug resistant mutations (DRMs) in the HCV proteins targeted by drugs greatly affect the treatment outcome and often lead to drug failure[2,4]. Numerous DRMs have been identified, corresponding to specific amino acid substitutions capable of negatively affecting the activity of DAAs either in vitro or in vivo in treated patients[4-7]. The extensive use of drugs for treating HCV places selective pressure that may lead to DRM-enriched viruses becoming prevalent in the population, which eventually would limit efficacy of the available drugs. Interestingly, many HCV DRMs are known to be individually deleterious for the virus[8-10]. Thus, it is important to understand the evolutionary factors facilitating the emergence of DRMs in HCV.

One such evolutionary factor is epistasis: a phenomenon in which the phenotypic effect of a mutation at one residue is dependent on mutations elsewhere in the protein sequence. Epistasis has been suggested to play an important role in HCV evolution[11-14]. In the case of HIV, a chronic-disease-causing RNA virus similar to HCV, drug resistance has been suggested to be mediated by epistasis[15-17], with the fitness cost incurred by DRMs compensated by other mutations in the drug-targeted protein[18-20]. For HCV, preliminary data suggests that DRMs may be involved in epistatic interactions[9,10,21]; however, a comprehensive understanding of the role played by epistasis in the evolution of DRMs in HCV is still lacking.

In this study, we investigate the role of pair-wise epistatic interactions in the evolution of drug resistance in the NS3 protein, one of the main targets of HCV drugs[22]. Using the globally prevalent HCV genotype 1a sequence data[23], we infer an in-silico model for the fitness landscape of HCV NS3, which takes into account the effect of both individual mutations and epistatic interactions between pairs of mutations. Our inferred model correlates strongly with multiple experimental data sources. Consistent with past studies on fitness landscape of HCV proteins[12-14,24], we find that epistatic interactions are important contributors to HCV fitness. Applying the fitness landscape model to study DRMs associated with NS3-targeting drugs (all known

[1]Department of Electronic and Computer Engineering, The Hong Kong University of Science and Technology, Clear Water Bay, Hong Kong SAR, China. [2]Department of Electrical and Electronic Engineering, University of Melbourne, Melbourne, VIC, Australia. [3]Department of Microbiology and Immunology, University of Melbourne, at The Peter Doherty Institute for Infection and Immunity, Melbourne, VIC, Australia. ✉e-mail: aaquadeer@connect.ust.hk; matthew.mckay@unimelb.edu.au

to be protease inhibitors[4]), we reveal that specific DRMs, referred as "SC-DRMs", are associated with strong compensatory epistatic interactions and are enriched in almost all NS3 drugs. We further show, by integrating the fitness landscape with an in-host evolutionary model, that under selective pressure from drugs it is relatively easy for HCV to incur SC-DRMs compared to other DRMs. In addition, we find that the number of SC-DRMs seems to negatively correlate with the efficacy of each NS3-specific drug. Overall, our results suggest an important role of epistasis in emergence of NS3-specific DRMs. Accounting for epistatic interactions might therefore be critical for studying resistance to current HCV NS3 drugs as well as for developing new drugs targeting the NS3 protein.

## Results

### Importance of epistatic interactions in predicting HCV NS3 fitness

To study the role of epistatic interactions in the development of drug resistance for HCV NS3, we first inferred a fitness landscape for the HCV NS3 protein using available sequences for genotype 1a. This inference involved determining a prevalence landscape—an estimate of the probability of observing an NS3 protein sequence among naturally occurring HCV populations—using a "least-biased" maximum entropy probabilistic model (Methods). In this model, the probability of observing a sequence $\mathbf{x} = [x_1, x_2, \ldots, x_N]$, is given by

$$P_{\mathbf{h,J}}(\mathbf{x}) = \frac{e^{-E_{\mathbf{h,J}}(\mathbf{x})}}{\sum_{\mathbf{x}'} e^{-E_{\mathbf{h,J}}(\mathbf{x}')}}, \text{ where } E_{\mathbf{h,J}}(\mathbf{x}) = \sum_{i=1}^{N-1} \sum_{j=i+1}^{N} J_{ij}\left(x_i, x_j\right) + \sum_{i=1}^{N} h_i(x_i),$$

(1)

where the parameters $h_i(x_i)$ represent the effect of mutations at individual residues $i$ and the parameters $J_{ij}\left(x_i, x_j\right)$ account for the effect of epistatic interactions between mutations at two different residues $i$ and $j$. $E_{\mathbf{h,J}}(\mathbf{x})$ represents the energy of sequence $\mathbf{x}$ which is inversely related to its prevalence. Similar models have been developed previously to predict direct residue contacts[25–27]. Here, we observed a strong negative correlation ($\bar{r} = -0.79$, Fig. 1; see "Methods" section for details) between the sequence energy predicted from the inferred model and the infectivity measurements for 45 sequences obtained from experimental studies[9,10,21,28,29] (listed in Supplementary Data 1). This suggests that the fitness landscape model serves as a good proxy for the intrinsic

fitness landscape of HCV NS3. This result was consistent with maximum-entropy-based fitness landscapes inferred in the past for other HCV proteins (NS5B[24] and E2[12,13]), and proteins from HIV[17,30–33] and poliovirus[34]. We also noted that the correlation obtained for the inferred model was much stronger than the correlation achieved by a model that ignores pair-wise epistatic interactions ($\bar{r} = -0.55$, Fig. 1 inset; see Methods for details). Based on a bootstrapping procedure, the difference in the correlation obtained for the two models was also found to be statistically robust (Supplementary Figure 1). This result suggests that considering epistatic interactions is important for reliably predicting the intrinsic fitness of HCV NS3.

### Association of DRMs with compensatory interactions

Many of the DRMs in HCV NS3 are known to be individually deleterious[8–10]. The emergence of DRMs may suggest that the fitness cost of DRMs are compensated by mutations elsewhere in the protein. To investigate if the residues involved in known NS3 DRMs[4–7] (listed in Table 1) were associated with compensatory interactions, we studied the model parameters $J_{ij}$. Large positive values of $J_{ij}$ in Eq. (1), that increase sequence energy and thereby decrease its predicted fitness, represent strong antagonistic interactions or negative epistasis between residues $i$ and $j$. Negative epistasis reduces the fitness of the double mutants and limits acquisition of additional mutations[35]. In contrast, large negative values of $J_{ij}$ in Eq. (1) represent strong compensatory interactions or positive epistasis between residues $i$ and $j$. Positive epistasis boosts the reproduction capability of double mutants, allowing viruses to acquire and retain drug resistance[36]. For HIV protease, positive epistasis among DRMs predicted by maximum entropy modeling (similar to ours) was shown to be consistent with the results of deep mutational scanning experiments[18]. In the case of HCV NS3, we found that pairs of mutations with large negative values of $J_{ij}$ were more likely to involve DRMs compared to random expectations (Supplementary Fig. 2). This suggests the enrichment of positive epistasis in residues associated with HCV NS3 DRMs.

Focusing on the top 10/100/300 pairs of mutations with large values of $-J_{ij}$, we observed that some specific DRMs were associated with particularly strong compensatory interactions (Fig. 2a). Henceforth, we refer to the DRMs enriched in the top 300 pairs ($p = 4.8 \times 10^{-61}$; two-sided Fisher's exact test) as strongly coupled DRMs, "SC-DRMs". The residues associated with the majority of SC-DRMs were involved in a sparse network of interactions (residues 36, 41, 54, 55, 71, 168, and 170), while a few had dense interaction networks (residues 80 and 122). Among the pairs involving SC-DRMs, two comprised both DRMs: residues 41 and 168 (ranked 1st), and residues 54 and 55 (ranked 3rd). The identified pairs involving SC-DRMs contain multiple residues that are in contact with the resolved NS3 protein structure, supporting the possibility that these residues may be interacting (Fig. 2b). Similar results have been reported for strongly

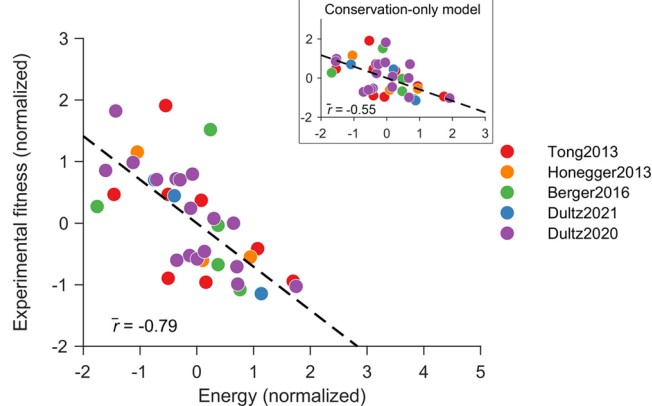

**Fig. 1 | Correlation between the sequence energy obtained from inferred NS3 fitness landscape and in-vitro infectivity measurements.** Normalized energy values computed from the inferred model correlate strongly with the experimental fitness measurements. In contrast, a conservation-only model provided a much lower correlation (inset). The legend shows references for fitness/infectivity measurements[9,10,21,28,29]. Normalization of fitness measurements and predicted model energies was performed by subtracting the mean from each data set and dividing by its standard deviation. Source data are provided as a Source Data file.

**Table 1 | List of NS3 drugs[4–7] and the associated DRMs**

| Drug | Class of DAAs | Residues involved in DRMs[a] |
|---|---|---|
| Telaprevir | NS3-specific | **36**, 43, **54**, 155, **170** |
| Vaniprevir | NS3-specific | **36**, 155, **168** |
| Boceprevir | NS3-specific | **36**, **41**, 43, **54**, **55**, 155, 158, **170** |
| Simeprevir | NS3-specific | **36**, 43, **54**, **80**, **122**, 138, 155, **168**, **170** |
| Danoprevir | NS3-specific | **41**, 43, 138, 155, **168** |
| Glecaprevir | Multi-protein | **36**, 56, **71**, **80**, 89, 155, **168**, **170** |
| Grazoprevir | Multi-protein | **36**, 43, **54**, **55**, 56, 77, **80**, 107, 109, **122**, 132, 155, 158, **168**, **170** |
| Voxilaprevir | Multi-protein | **36**, **41**, 43, **54**, **55**, 56, **80**, **122**, 155, **168**, **170**, 180 |
| Paritaprevir | Multi-protein | **36**, 43, 56, **80**, **122**, 155, **168** |

[a]Residues associated SC-DRMs are shown in bold.

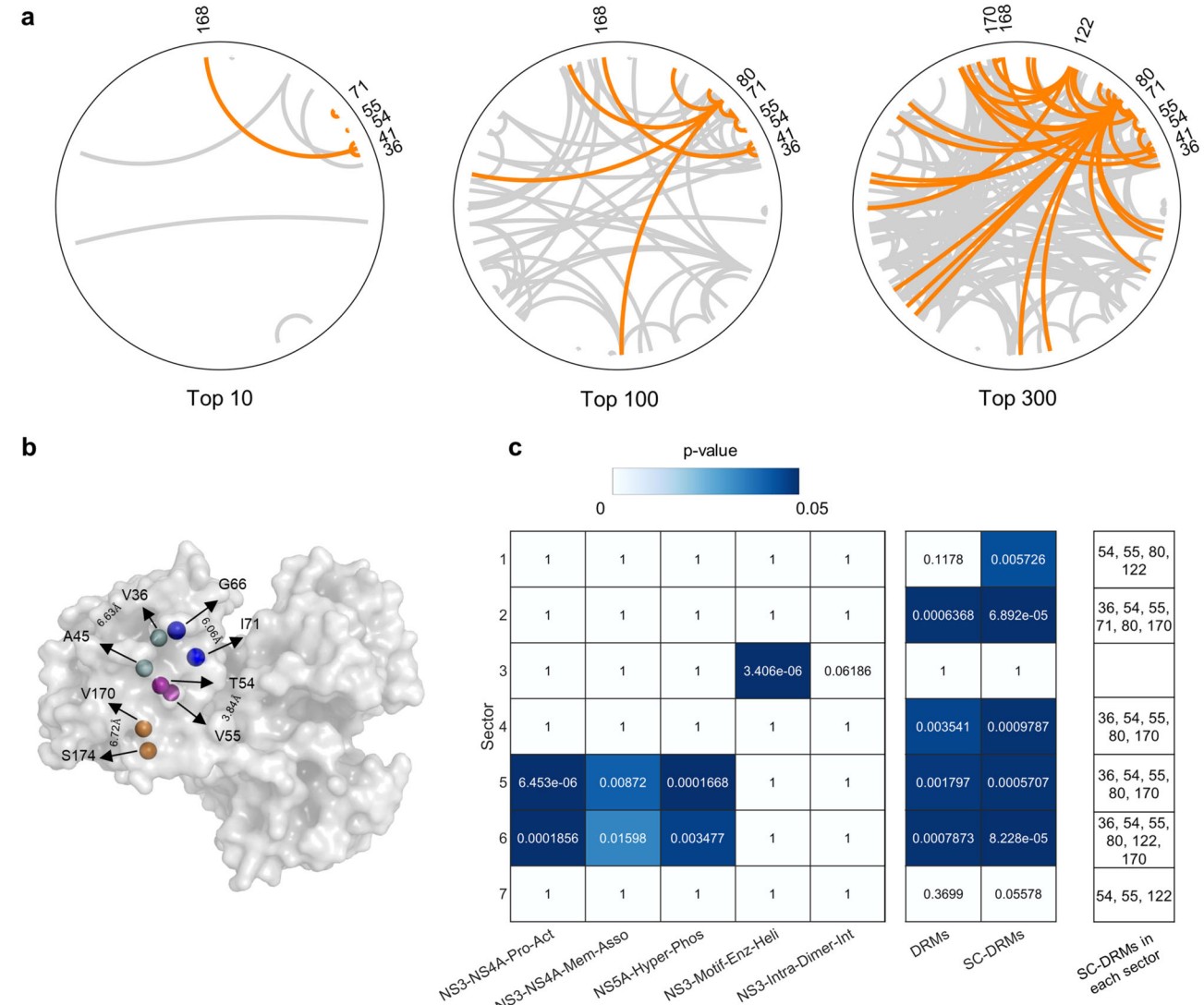

**Fig. 2 | Identification of SC-DRMs and their significance. a** Network of interactions between top 10/100/300 ranked mutations (ranked by the values of $-J$ from Eq. (1)). Interactions linking at least one DRM are shown orange, and links between non-DRMs are shaded in gray. **b** Pairs of interacting residues involving SC-DRMs that are in contact based on the crystal structure of the NS3 protein (PDB ID: 4B6E [https://doi.org/10.2210/pdb4b6e/pdb]). The carbon-alpha atoms of each pair of residues are shown as colored spheres, and the distance between each pair is also labeled. Two residues were assumed to be in contact if their carbon-alpha atoms were <8 Å apart. **c** Inferred NS3 sectors and their association with SC-DRMs. Sectors (listed in Supplementary Table 1) were inferred using the GUI implementation of

the robust co-evolutionary analysis approach, RocaSec[37,90]. The statistical significance (p-values) was determined using one-sided Fisher's exact test. In addition to the set of residues associated with DRMs and SC-DRMs, the following known NS3 biochemical regions were provided to the Rocasec (listed in Supplementary Table 2). (i) NS3-NS4A-Pro-Act NS3-NS4A interface for protease activation[40]; (ii) NS3-NS4A-Mem-Asso: NS3-NS4A membrane association[41]; (iii) NS5A-Hyper-Phos: NS5A hyper-phosphorylation[42,43]; (iv) NS3-Motif-Enz-Heli: motif important for enzymatic and helicase activities in NS3[44]; and (v) NS3-Intra-Dimer-Int: intra-dimer interface in NS3 helicase[45]. Source data are provided as a Source Data file.

coupled pairs of residues in maximum entropy models for multiple protein families[25].

To explore if the SC-DRMs identified by our model are also associated with groups of residues known to mediate different NS3 functions, we applied a robust co-evolutionary analysis approach that we developed previously[37]. Distinct from maximum-entropy-based fitness landscape models, this approach identifies collective groups of co-evolving residues (called sectors), rather than pair-wise interactions. Such sectors, for HIV and HCV, have been shown to distinctly associate with protein functional or structural domains[37–39]. Applying the co-evolutionary analysis on the NS3 data considered in this work, we found that SC-DRMs were enriched in multiple inferred sectors (Fig. 2c; for details of inferred sectors, see Supplementary Table 1). (A similar enrichment to SC-DRMs was noted for DRMs but with slightly weaker statistical significance.) Of the sectors with known biochemical

associations[40–45] (for details of the experimentally-defined biochemical domains, see Supplementary Table 2), SC-DRMs were particularly enriched in the sectors linked with the NS3-NS4A interface known to mediate serine protease activity of NS3[40]. While there is no overlap between the SC-DRM-associated residues and the residues known to be critical for NS3 protease activity, inference of a sector encompassing both of these sets of residues suggests that they may be co-evolving. Interestingly, SC-DRMs were not enriched in the inferred sector associated with the NS3 helicase activity. This is in line with the fact that none of the approved HCV NS3 drugs are helicase inhibitors[4].

**Model predictions correlate with known NS3 DRM compensation data**

Experimental data derived from in vivo or in vitro studies offers the most direct evidence for compensatory mutations associated with SC-

**a**

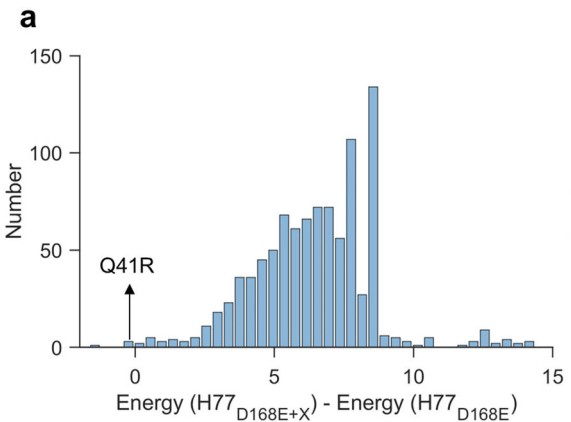
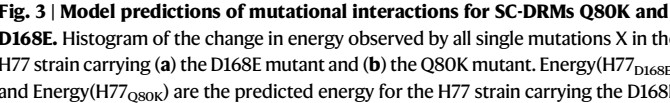

**b**

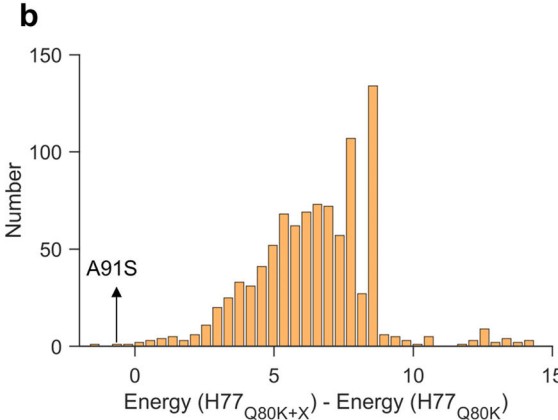

**Fig. 3 | Model predictions of mutational interactions for SC-DRMs Q80K and D168E.** Histogram of the change in energy observed by all single mutations X in the H77 strain carrying (**a**) the D168E mutant and (**b**) the Q80K mutant. Energy(H77$_{D168E}$) and Energy(H77$_{Q80K}$) are the predicted energy for the H77 strain carrying the D168E and Q80K mutant. The predicted energy for the H77 strain carrying the D168E mutant and an additional single mutation X, as well as for the H77 strain carrying the Q80K mutant and an additional single mutation X, are given by Energy(H77$_{D168E+X}$) and Energy(H77$_{Q80K+X}$), respectively. Source data are provided as a Source Data file.

DRMs. However, such data is currently limited. In vivo evidence is available for the SC-DRM Q80K which has been reported to co-occur with the A91S mutation among individuals who experience HCV treatment failure[10]. This compensatory interaction has also been observed in vitro in the H77 strain[10]. Experimental evidence of SC-DRM D168E being compensated by Q41R[9] has also been reported for the H77 strain. These two pairs of compensatory mutations were both associated with large values of $-J_{ij}$; Q80K and A91S were ranked 60th, and D168E and Q41R were ranked 1st (Fig. 2a).

We further investigated the mutational interactions predicted by our model for these two SC-DRMs, Q80K and D168E. We specifically examined the energy changes in the H77 strain bearing the D168E or Q80K mutants (denoted H77$_{D168E}$ and H77$_{Q80K}$ respectively) upon introducing all possible mutations. A negative energy change indicates increased fitness, whereas positive change suggests a fitness reduction. Strikingly, our model predicted that the Q41R and A91S mutations yielded the second-most negative energy change compared to all other mutations in the H77$_{D168E}$ and H77$_{Q80K}$ strains, respectively (Fig. 3). This outcome is consistent with the documented compensatory roles of these mutations for DRMs D168E and Q80K, and points to the specificity of our model in describing epistatic compensatory pathways.

Extending the analysis to predict compensatory mutations associated with SC-DRMs in different sequence backgrounds as opposed to only H77 (see "Methods" section for details), we found that 168E and 41R were compensatory (for each other) for all sequence backgrounds, while 91S compensated for 80K in ~23% of sequence backgrounds (Supplementary Table 3). We also identified potential compensatory mutations for SC-DRMs 36L, 55A, 122C/G, and 177V. These identify specific targets for future experimental studies.

**Enrichment of SC-DRMs in NS3 drugs**

Currently, there are nine known NS3-targeting drugs used for treating HCV genotype 1a infections[4–7]. These drugs can be divided into two classes: NS3-specific DAAs that exclusively target NS3 (telaprevir, boceprevir, simeprevir, vaniprevir, and danoprevir) and multi-protein DAAs that target NS3 together with other proteins (paritaprevir, grazoprevir, voxilaprevir, and glecaprevir). For all of these drugs, a total of 20 NS3-specific DRMs have been identified, ranging from 3 to 15 DRMs per drug. Each NS3 drug, irrespective of the drug class, was found to comprise at least two identified SC-DRMs (Table 1). This association reached statistical significance ("Methods" section) for most drugs (5/9; two from NS3-specific DAAs and three from multi-protein DAAs;

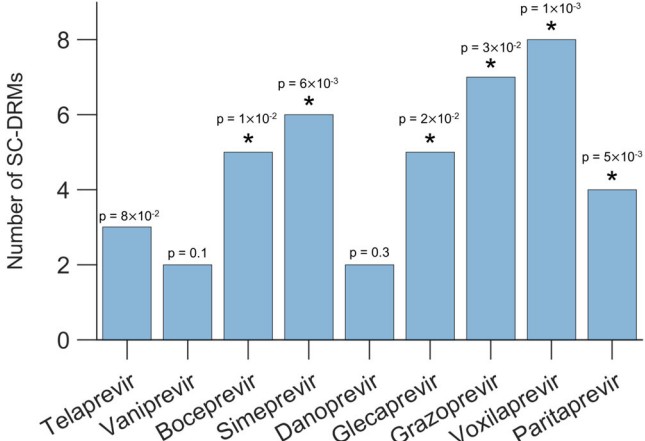

**Fig. 4 | SC-DRMs are enriched in NS3 drugs.** Statistical significance of the identified number of SC-DRMs associated with each drug. The *p*-value measures the probability of observing by a random chance at least the observed number of SC-DRMs among all DRMs associated with a drug (one-sided test; see "Methods" section for details). Results with *p*-value < 0.05 are marked with a star on the top of each bar. Source data and exact *p*-values are provided as a Source Data file.

Fig. 4) and was robust to the number of top-coupled pairs of mutations used for defining SC-DRMs (Supplementary Fig. 3). In contrast, the remaining DRMs (non-SC-DRMs) were generally not significantly enriched in drugs (1/9, Supplementary Fig. 4). The enrichment of the identified SC-DRMs in NS3-targeting DAAs suggests that they play a significant role in conferring resistance to both classes of drugs. This observation suggests that epistasis is an important factor contributing to the acquisition of resistance to NS3 drugs.

Some DRMs have been reported to disrupt drug binding while having minimum effect on the NS3 protease function[7,46]. Thus, we investigated whether SC-DRMs may also be enriched in binding residues of the drugs. Structures for four drugs in complex with the NS3 protein are available (PDB ID: [3M5L] for danoprevir, [3SU3] for vaniprevir, [3SV6] for telaprevir, and [3SUD] for grazoprevir). Based on these, we identified binding residues for each drug as those NS3 residues that are within 5Å of a drug atom[46]. While, for each drug, not all drug-specific DRMs are located within the binding residues (Fig. 5a), DRMs were found to be statistically significantly enriched within them (Fig. 5b). The same was also true for SC-DRMs of danoprevir and

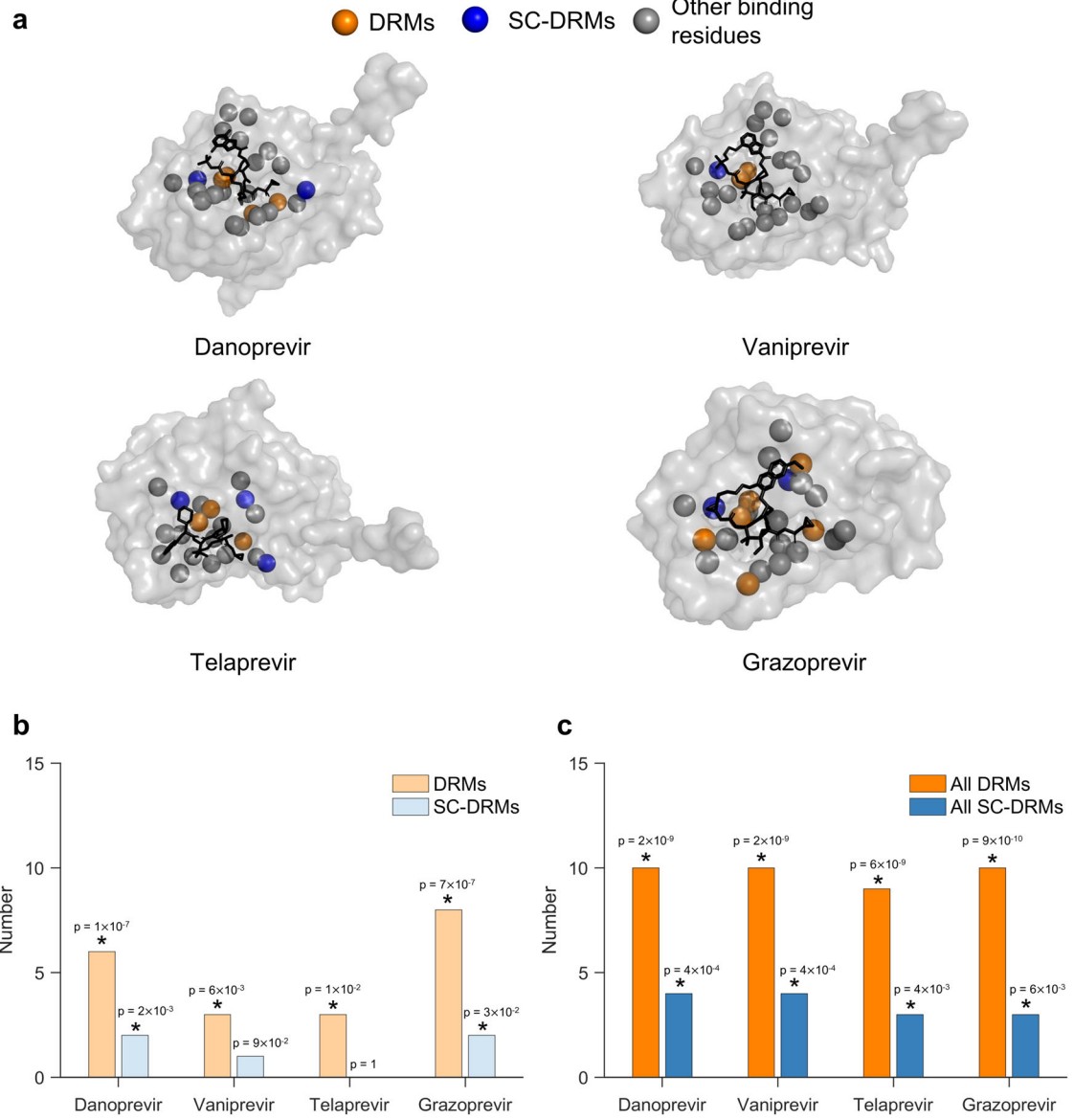

**Fig. 5 | SC-DRMs appear to impact binding of NS3 drugs through direct interactions. a** Binding residues of drugs shown on the crystal structure of the NS3 protein-drug complexes (PDB ID: 3M5L [https://doi.org/10.2210/pdb3m5l/pdb] for danoprevir, 3SU3 [https://doi.org/10.2210/pdb3su3/pdb] for vaniprevir, and 3SV6 [https://doi.org/10.2210/pdb3sv6/pdb] for telaprevir and 3SUD [https://doi.org/10.2210/pdb3sud/pdb] for grazoprevir). The carbon-alpha atoms of the identified drug-binding residues are shown in colored spheres. The drug-binding residues associated with DRMs and SC-DRMs for each drug are shown in green and blue, respectively, while those that do no fall under DRMs are shown in gray. Drugs in each structure are shown as black sticks. The NS3 residues within 5 Å7D2of drug atoms were considered as drug-binding residues. **b, c** Statistical significance of the number of (**b**) drug-specific DRMs/SC-DRMs and (**c**) all DRMs/SC-DRMs in binding residues of each of the four considered drugs. Here, drug-specific DRMs/SC-DRMs are listed in Table 1 for each of the four drugs, while all DRMs/SC-DRMs refer to the DRMs/SC-DRMs known for all drugs. The *p*-value measures the probability of observing by a random chance at least the observed number of DRMs or SC-DRMs among all binding residues for each drug (one-sided test; see "Methods" section for details). Results with *p*-value < 0.05 are marked with a star on the top of each bar. Source data and exact *p*-values are provided as a Source Data file.

grazoprevir, but not for vaniprevir and telaprevir (Fig. 5b). This suggests that DRMs associated with these drugs, and SC-DRMs in the case of danoprevir and grazoprevir, may confer resistance by directly affecting drug binding.

The known DRMs for each drug have been identified either via limited in-vitro experiments based on their adverse impact on DAA activity, or in-vivo in a few treated patients in clinical trials[4]. Hence, information of DRMs available for each NS3 drug may not be complete. This, in addition to the observation that several DRMs are shared across multiple NS3 drugs (Table 1), motivates analysis of the enrichment of the collective set of DRMs (i.e., associated with all NS3 drugs) in the binding residues of the four drugs with available structures

(detailed in Supplementary Table 4). In this case, the enrichment of DRMs was statistically more significant (Fig. 5c) than that observed for drug-specific DRMs (Fig. 5b). SC-DRMs were also now statistically significantly enriched in all four drugs (Fig. 5c). This analysis identified numerous DRMs and SC-DRMs that have been determined for specific drugs and which lie within the binding footprints of other drugs, but have not yet been reported as conferring resistance for those drugs. Hence, these may correspond to putative DRMs or SC-DRMs that have yet to be observed in-vitro or in-vivo. SC-DRMs at residues 41, 55, and 168 were common among the binding residues of all four drugs, with two of these SC-DRMs (41 and 168) known to be involved in compensatory interactions via ex-vivo experiments[9]. Collectively, this analysis

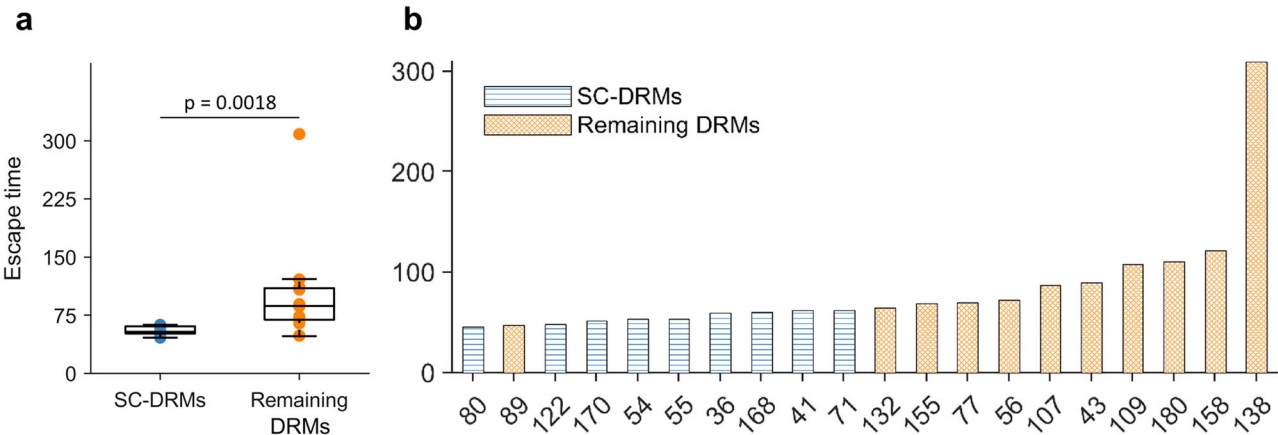

**Fig. 6 | Escape time of residues involved in NS3 DRMs. a** Comparison between escape time of residues involved in SC-DRMs and the remaining residues involved in DRMs. In each box plot, the middle line indicates the median, the edges of the box represent the first and third quartiles, and whiskers extend to span a 1.5 interquartile range from the edges. The reported *p*-value was calculated using the two-sided Mann–Whitney test ($n_1 = 9$ SC-DRMs and $n_2 = 11$ remaining DRMs). **b** Individual escape time of residues involved in DRMs of the NS3 protein. SC-DRMs are shown in blue and the remaining DRMs in orange. Source data are provided as a Source Data file.

suggests that DRMs may facilitate resistance by interrupting binding ability of drugs to the NS3 protease. In the case of SC-DRMs, this resistance is further facilitated through compensatory interactions of networked mutations, wherein the deleterious effect of a SC-DRM might be compensated by another mutation in the network.

Our analysis can be used to predict mutations that could potentially confer drug resistance. Specifically, we identified 25 binding residues for drugs with known structures (PDB ID: [3M5L] for danoprevir, [3SU3] for vaniprevir, [3SV6] for telaprevir, and [3SUD] for grazoprevir), out of which 14 residues were not previously associated with any known DRMs (Supplementary Table 5). However, based on our model, we found that mutations at four of these 14 residues (residues 78, 79, 123, and 159) were associated with strong compensatory interactions. Furthermore, at least two of these four residues were present in the binding residues of all four drugs considered, suggesting that mutations at these residues may potentially confer resistance to the drugs.

### SC-DRMs provide an easy escape from drug-induced selection pressure

In general, the initiation of drug treatment alters the in-host environment in which HCV replicates. Selective pressure exerted by a drug may promote mutations that simultaneously resist the drug and maintain replicative capacity. Since we observed that SC-DRMs were statistically significantly enriched in most drugs (Fig. 4) while the remaining DRMs (non-SC-DRMs) were generally not (Supplementary Figure 4), we investigated whether it is easier for these SC-DRMs to accumulate in the viral population than other DRMs. We integrated the inferred fitness landscape in an in-host evolutionary model to quantify the average time, termed "escape time", that the virus takes to escape from selective pressure targeting the residues involved in DRMs (see Methods for details). This Wright-Fisher-like model[47] accounts for the complex stochastic dynamics involved in in-host evolution of HCV quasispecies, including host-virus and virus-virus interactions, and multiple pathways that HCV may employ to escape from selective pressure exerted by a drug. Similar evolutionary models have been employed previously by us and others for determining the average immune escape time associated with residues in HCV E2[12,13] and HIV Gag[32].

Contrasting the escape times of residues associated with SC-DRMs against those for the residues associated with the remaining NS3 DRMs revealed that the former set of residues carries shorter escape times ($p = 0.0018$, Mann-Whitney test; Fig. 6a). Investigating the escape time of the residues associated with individual NS3 DRMs showed that almost all residues associated with SC-DRMs had a shorter escape time than the remaining NS3 DRMs (Fig. 6b). These results suggest that SC-DRMs provide relatively easy pathways, enabled via epistatic interactions, for HCV to escape drug-induced selective pressure. This provides a rationale for the enrichment of SC-DRMs in NS3 drugs (Fig. 4).

### Accumulation of SC-DRMs appears to impact the efficacy of NS3 drugs

We further explored whether there exists any (inverse) relation between the number of SC-DRMs associated with a drug and the expected efficacy of the drug. Efficacy data was collected from multiple clinical studies (listed in Supplementary Table 6[48–69]). (Multi-protein DAAs have generally been reported to achieve much higher efficacy than NS3-specific DAAs.) We observed a strong negative correlation between the number of SC-DRMs and the efficacy of NS3-specific DAAs ($r = -0.67$; Fig. 7a) as well as for multi-protein DAAs ($r = -0.77$; Fig. 7b). While the limited number of drugs didn't allow these tests to reach statistical significance, the observed strong negative correlation is suggestive of the potential impact that abundance of SC-DRMs can have on the efficacy of HCV drugs.

## Discussion

Emergence of DRMs is a common phenomenon observed in patients undergoing HCV drug therapy, which often negatively impacts the treatment outcome. Studies indicate that a notable proportion of patients who fail DAA treatment have DRMs, with prevalence rates ranging from 20-90% depending on the specific DAA used[70–72]. The accumulation of DRMs in such patients carries clinical and public health concern due to the limited treatment options available[73] and the potential transmission of drug-resistant strains to other individuals[74]. In addition, the widespread use of HCV DAAs could lead to the prevalence and dominance of DRMs in the future, similar to the observed increase in HIV DRMs with the use of antiretroviral therapy (ART)[75]. Thus, it is important to understand the evolutionary factors that contribute to emergence of HCV DRMs.

Many of the DRMs are known to be individually deleterious. Therefore, other evolutionary factors, such as epistasis, may be facilitating their emergence. To investigate this aspect, we first inferred a fitness landscape for the NS3 protein (one of the main proteins

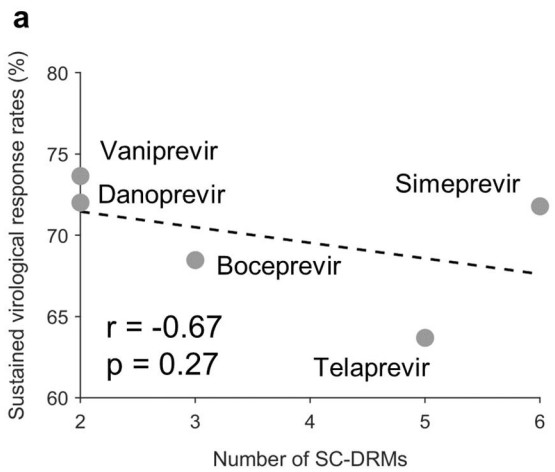
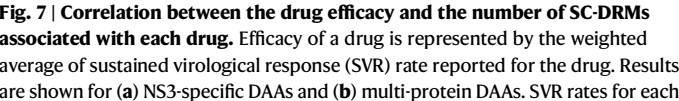
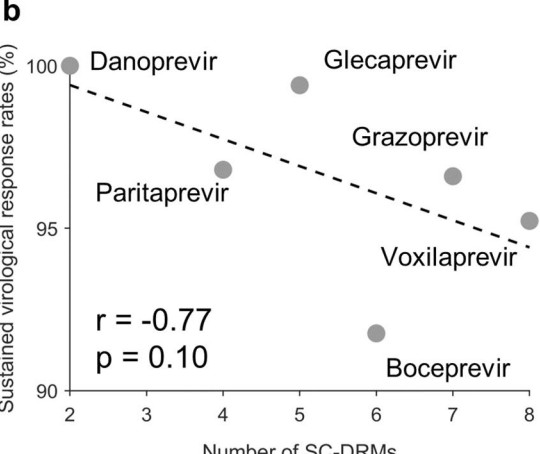

**Fig. 7 | Correlation between the drug efficacy and the number of SC-DRMs associated with each drug.** Efficacy of a drug is represented by the weighted average of sustained virological response (SVR) rate reported for the drug. Results are shown for (**a**) NS3-specific DAAs and (**b**) multi-protein DAAs. SVR rates for each drug were curated from the literature[48,69] (listed in Supplementary Data 1; see Supplementary Table 6 for details). Source data are provided as a Source Data file. The *p*-values measure the two-sided significance level of the Spearman correlation.

targeted by HCV drugs) considering both the effects of individual mutations and interactions between mutations at different residues. Predictions obtained from the inferred model correlated well with multiple experimental data sources. The analysis of model parameters capturing pair-wise epistatic interactions (couplings) showed that certain DRMs, namely SC-DRMs, were associated with strong compensatory interactions, were seemingly involved in mediating protease function, and were prevalent among NS3 drugs. Upon integrating the inferred fitness landscape into an evolutionary model, we found that SC-DRMs were associated with a relatively easy escape from drug-induced selection pressure. The number of SC-DRMs also appeared to correlate inversely with the efficacy of NS3-targeting drugs. Overall, our results suggest that epistatic interactions associated with SC-DRMs provide easy pathways that contribute to drug resistance.

The inference of the fitness landscape from population-level sequence data for NS3 is complex given the selective pressure from host immune responses and recent use of DAAs. The high correlation between prevalence and fitness of NS3 (Fig. 1) may be surprising, but a similar relationship has been reported for other HCV proteins[12,13,24], as well as several HIV proteins[17,30–33]. The mechanistic rationale for this correspondence has been previously proposed for HIV proteins, with three key factors identified[76]: (i) a diverse and largely ineffective immune response due to host genetic diversity, (ii) reversion to the ancestral (fitter) sequence upon transmission to a new host, and (iii) the absence of robust and effective natural or vaccine-induced herd memory responses, which would shift the virus away from the steady state. Although HCV differs from HIV, it shares several similarities and may also involve these factors. Specifically, since the lack of a functional vaccine, most NS3 sequences sampled from chronic patients, and NS3 being a target of T cells[77], it is likely that NS3 experiences diverse and ineffective immune responses in such patients. Reversion to the consensus amino acid upon HCV transmission to a new host has also been documented[78].

DAAs could potentially lead to population-wise selective pressure that may bias our data, however such effects are not expected to be strong. This is because DAAs are currently only available to a limited fraction of HCV-infected individuals (less than 20%[79]). To examine this more explicitly for the NS3 data set that we used (comprising 7370 sequences), we investigated the 58 papers from which these sequences were reported. This analysis revealed that the large majority of the sequences (5877 sequences) were indeed from drug-naïve patients. Comparing statistical properties of the complete dataset

(7370 sequences) with those of the drug-naïve subset (5877 sequences) revealed a strong correlation ($r > 0.9$, Supplementary Fig. S5a) between the mutation frequencies and pair-wise mutation frequencies in both datasets. We also constructed a maximum entropy model using only the drug-naïve sequences and found that the predicted sequence energies from the drug-naïve model exhibited a correlation ($r = -0.70$, Supplementary Fig. S5b) with the in-vitro fitness measurements, which was comparable to the correlation observed with the complete dataset.

It is noteworthy that 36 out of the 45 fitness measurements compiled from different experimental studies were associated with DRMs. The high correlation between our model's predictions (inferred using the complete dataset) and the experimental fitness measurements (Fig. 1) supports that our model can accurately capture the intrinsic effect of DRMs, despite being trained mainly on drug-naïve sequences. This is because DRMs have been observed in drug-naïve patients as well[80,81]. We further evaluated the correlation between our model predictions and the 36 fitness measurements exclusively associated with DRMs and found it also to be high ($r = -0.72$, Supplementary Fig. 6), providing additional evidence for the ability of our model to capture the effect of NS3 DRMs.

Among the pairs of residues involving SC-DRMs (Fig. 2a), two pairs (pairs 41–168 and 80–91) have been demonstrated to be involved in compensatory interactions using ex-vivo experiments[9,10]. For the remaining pairs, based on the available NS3 protein structure, a few pairs were found to be in contact (pairs 36–45, 54–55, 66–71, and 170–174) (Fig. 2b), further suggesting that epistatic interactions may exist between these pairs. Such pairs of residues involving SC-DRMs provide directions for future experimental studies investigating compensatory interactions. These may include experiments that quantify the change in replicative fitness, protein folding, or protease enzymatic function upon mutating these pairs of residues individually and simultaneously.

Our analysis revealed an inverse correlation between the number of SC-DRMs and the efficacy of NS3-targeting drugs (Fig. 7). However, this relationship may be influenced by various confounding factors. These include, for instance, differences in drug dosage and duration, whether the drug was used in combination with interferon and/or ribavirin, whether peg-interferon therapy was administered prior to DAA treatment, as well as different host-specific factors of patients (cirrhotic status, HCV RNA level, and HLA composition). These factors could not be explicitly accounted for in our analysis. Further research

with more detailed data on these confounding factors is needed to fully understand their influence on drug efficacy and to provide more robust conclusions.

While we focused on epistasis within the NS3 protein, inter-protein interactions may also play a role in conferring resistance to drugs. For instance, interactions between NS5A and NS5B – the two other proteins targeted by multi-protein DAAs – are known to be critical for HCV RNA replication[82], and hence, interactions between these proteins might also affect the emergence of DRMs. Moreover, we observed a marginally significant ($p = 0.07$, Mann–Whitney test) difference in the number of DRMs of NS3-specific DAAs (4–10) and multi-protein DAAs (8–16). This suggests that the resistance profile of these two classes of drugs may be different, thereby motivating further investigation into inter-protein interactions between different HCV proteins for conferring drug resistance.

We predicted SC-DRMs to be associated with shorter escape time from drug-induced selective pressure (Fig. 6a), which provides support to their enrichment in each NS3 drug (Fig. 4). In addition, this analysis revealed that DRMs at residues 138 and 158 were associated with much higher escape time compared to other DRMs (Fig. 6b). This is suggestive that it may be hard for HCV to escape from drug-induced selective pressure by incurring mutations at these NS3 residues. Thus, preferentially targeting such residues may be desirable for designing robust next-generation HCV drugs and vaccines.

The high genetic variability among different HCV genotypes and subtypes makes them differently susceptible to the development of NS3 DRMs[83]. Thus, certain drugs only work for specific genotypes/subtypes. For instance, telaprevir, boceprevir, and vaniprevir only work for genotype 1 infections, while simeprevir, paritaprevir, and danoprevir can be used for treating both genotypes 1 and 4 infections[4]. In line with this, we have also shown in a previous study that evolutionary constraints are different across HCV subtypes[13], which may also be a contributing factor in the observed difference in efficacy of drugs against different genotypes/subtypes. Thus, while we focused here only on NS3 subtype 1a, extending this analysis to study different HCV genotypes/subtypes might be helpful to understand genotype-specific differences in drug efficacies.

There are also multiple limitations of our study. First, we focused on pair-wise epistatic interactions only. Higher-order epistatic interactions may also contribute to viral fitness, and these are not captured by our model. Inferring such higher-order effects is challenging and requires larger data sets with higher sequence variability. Second, we are unable to systematically study the resistance profile of multi-protein DAAs. This would require a joint model considering multiple proteins together. Again, data limitations currently preclude the development of such joint multi-protein models. Third, while our analysis shows an inverse relation between the efficacy of a given drug and the number of associated SC-DRMs, there are multiple factors that may potentially confound a relative analysis of reported drug efficacies. These include differences among drugs with respect to dosage, duration, and administration, along with differences among characteristics of patients. Each of these factors may influence the efficacy of drugs, and they could not be explicitly accounted for in our analysis. More detailed data related to the effect of these confounding factors is required to deconvolve the impact of SC-DRMs on drug efficacy.

## Methods

### Sequence data preprocessing
We downloaded 9683 NS3 genotype 1a aligned protein sequences (coverage ≥ 99%) from the HCV-GLUE database, http://hcv.glue.cvr.ac.uk[5,6]. We conducted principal component analysis (PCA) of the pairwise similarity matrix (9683 × 9683) constructed from the sequence data[84] to remove 148 outlying sequences. Briefly, all those sequences were considered outliers that appeared at more than 3 scaled median absolute deviations away from the median of either the first or second PC[85]. The scaled median absolute deviations is given by: $c \times$ median $(\text{abs}(A_i - \text{median}(A)))$, where $A$ is the first or second PC, $A_i$ is the $i$th element in the first or second PC , $c = -1/(\sqrt{2} \times \text{erfcinv}(3/2)) \approx 1.482$, and erfcinv() is the inverse complementary error function. To avoid unnecessary patient bias that can compromise model predictive ability (Supplementary Figure 7), we excluded 2167 sequences that were not associated with any patients. These filtering procedures resulted in $M = 7370$ sequences (accession numbers listed in Supplementary Data 2) from $W = 4773$ patients. Next, we excluded from this data 116 fully conserved residues, i.e., residues where no mutation was observed in any sequence. This excluded residue 156 from our analysis as it was fully conserved in our data, and therefore, DRMs associated with it were not investigated in our work. The final multiple sequence alignment (MSA) comprised $M = 7370$ sequences and $N = 515$ residues.

### Inference of HCV NS3 fitness landscape
We constructed a least-biased maximum entropy model for the NS3 protein that can reproduce the single and double mutant probabilities of the MSA that are given by

$$f_i(a) = \frac{1}{W} \sum_{k=1}^{M} w_k \delta(x_i^k, a)$$
$$f_{ij}(a, b) = \frac{1}{W} \sum_{k=1}^{M} w_k \delta(x_i^k, a) \delta(x_j^k, b). \tag{2}$$

Here, $x_i^k$ is the $i$th amino acid in the sequence $k$, $w_k$ is the reciprocal of the number of MSA sequences from the patient that sequence $k$ was obtained from, and $\delta(a, b)$ is the Kronecker delta function. As described in Eq. (1), the maximum entropy model assigns a sequence $\mathbf{x} = [x_1, x_2, \ldots, x_N]$ the probability

$$P_{\mathbf{h},\mathbf{J}}(\mathbf{x}) = \frac{e^{-E_{\mathbf{h},\mathbf{J}}(\mathbf{x})}}{Z}, \text{ where } E_{\mathbf{h},\mathbf{J}}(\mathbf{x}) = \sum_{i=1}^{N-1} \sum_{j=i+1}^{N} J_{ij}(x_i, x_j) + \sum_{i=1}^{N} h_i(x_i),$$

where $\mathbf{h}$ is the set of all fields that represent the effect of mutations at a single residue, and $\mathbf{J}$ is the set of all couplings that represent the effect of interactions between mutations at two different residues. $Z = \sum_{\mathbf{x}} e^{-E_{\mathbf{h},\mathbf{J}}(\mathbf{x})}$ is a normalization factor, and $E_{\mathbf{h},\mathbf{J}}(\mathbf{x})$ represents the energy of sequence $\mathbf{x}$. The fields $\mathbf{h}$ and couplings $\mathbf{J}$ are chosen such that the single and double mutant probabilities obtained from the model match respectively $f_i(a)$ and $f_{ij}(a, b)$ (Eq. (2)), i.e.,

$$\sum_{\mathbf{x}} \delta(x_i, a) P_{\mathbf{h},\mathbf{J}}(\mathbf{x}) = f_i(a)$$
$$\sum_{\mathbf{x}} \delta(x_i, a) \delta(x_j, b) P_{\mathbf{h},\mathbf{J}}(\mathbf{x}) = f_{ij}(a, b). \tag{3}$$

The problem of inferring the model parameters can be cast as the following convex optimization problem[24]

$$\left(\mathbf{h}^*, \mathbf{J}^*\right) = \frac{\arg\min}{\mathbf{h},\mathbf{J}} \text{ KL}\left(P_0 || P_{\mathbf{h},\mathbf{J}}\right) = \frac{\arg\min}{\mathbf{h},\mathbf{J}} \sum_{\mathbf{x}} P_0(\mathbf{x}) \ln \frac{P_0(\mathbf{x})}{P_{\mathbf{h},\mathbf{J}}(\mathbf{x})} \tag{4}$$

where KL$(\cdot || \cdot)$ denotes the Kullback-Leibler divergence between probability distributions, and

$$P_0(\mathbf{x}) = \frac{1}{W} \sum_{k=1}^{M} w_k \delta\left(\mathbf{x}^k, \mathbf{x}\right)$$

is the patient-weighted probability of observing strain $\mathbf{x}$ in the MSA.

To obtain the fields $\mathbf{h}$ and couplings $\mathbf{J}$ such that the inferred model reproduces the single and double mutant probabilities of the MSA, we used the GUI realization of MPF-BML[86], an efficient inference

framework introduced in ref. 33. This framework has been used previously to infer the fitness landscape of the HIV envelope protein[33] and the HCV E2 protein[12,13]. The MPF-BML inference framework comprises the following three steps:

1. The first step in the inference framework is to prevent overfitting of our model and reduce the computational time. To achieve this, we employ a process that retains only the top $k_i$ most frequent mutants out of the total $q_i$ mutations. The remaining mutants, $q_i - k_i$ in number, are grouped together in a way that the entropy associated with the grouping accounts for a certain fraction $\phi$ of the entropy without grouping. For a specific residue $i$, we need to find the smallest integer value of $k_i$ that satisfies the following condition:

$$S_i(k_i) \geq \phi S_i(q_i),$$

where

$$S_i(k_i) = -\sum_{a=1}^{k_i} f_i(a) \ln f_i(a) - \bar{f}_i \ln \bar{f}_i,$$

and

$$\bar{f}_i = \sum_{a=k_i+1}^{q_i} f_i(a).$$

$\phi$ is chosen such that the mean of

$$\beta_i(\phi) = \frac{\sum_{a=1}^{q_i} \left( f_i(a) - \bar{f}_i(a) \right)^2}{\sum_{a=1}^{q_i} \frac{f_i(a)(1-f_i(a))}{W}}$$

is approximately one, and

$$\bar{f}_i(a) = \begin{cases} f_i(a) & \text{if } a < k_i + 1 \\ \bar{f}_i & \text{if } a = k_i + 1 \\ 0 & \text{if } a > k_i + 1 \end{cases}.$$

The main concept is to achieve equilibrium between the bias (numerator) and the variability in the estimated amino acid frequencies (denominator) until they become approximately equal. Specifically, each amino acid at the $i$th residue is encoded using $q_i$ binary digits, where $q_i = k_i + 1$, and $k_i$ represents the number of mutants after combining. The $j$th most frequent amino acid is then represented by a $q_i$-bit binary code with the value of $2^{j-1}$. Consequently, the consensus sequence is represented by an all-zero vector. We define a binary matrix based on the amino acid matrix as **Y**, with the $i$th row denoted by $\mathbf{y}_i$.

2. Because the normalization factor $Z$ is intractable in Eq. (1), the second step, called the minimum probability flow (MPF) method, is to alleviate this computational burden by replacing $P_{\mathbf{h},\mathbf{J}}(\mathbf{x})$ with an alternate probability mass function (PMF) by considering a continuous-time Markov chain whose states correspond to the $B = \prod_{i=1}^{N}(q_i + 1)$ possible sequences. The master equation describing this Markov chain is given by

$$\frac{d}{dt}P_t(\mathbf{y}_i|\mathbf{h},\mathbf{J})) = \sum_{j=1, j \neq i}^{M} \Gamma_{ij}P_t(\mathbf{y}_j|\mathbf{h},\mathbf{J}) - \sum_{j=1, j \neq i}^{M} \Gamma_{ji}P_t(\mathbf{y}_i|\mathbf{h},\mathbf{J})), \quad (5)$$

where $P_t(\mathbf{y}_i|\mathbf{h},\mathbf{J}))$ denotes the probability of observing $\mathbf{y}_i$ at time $t$, and when $t = 0$, we have $P_t(\mathbf{h},\mathbf{J}) = P_0$. We can derive the solution to Eq. (5) as

$$P_t(\mathbf{y}_i|\mathbf{h},\mathbf{J})) = \left[ \exp(t\Gamma)P_0 \right]_i,$$

where $[\mathbf{a}]_i$ denotes the $i$th element of the vector $\mathbf{a}$. The matrix $\Gamma$ is the $B \times B$ transition rate matrix with $(i,j)$th element $\Gamma_{ij}$ given such that

$$\lim_{t \to \infty} P_t(\mathbf{y}_i|\mathbf{h},\mathbf{J})) = P(\mathbf{y}_i|\mathbf{h},\mathbf{J}).$$

The details of matrix $\Gamma$ can be found in ref. 87. The idea is that, regardless of the initial values of **h** and **J**, the PMF can evolve towards $P_t(\mathbf{h},\mathbf{J})$ as time increases. Then we choose a $t$ to make this problem tractable. After replacement, Eq. (4) expands as a Taylor series around $t = 0$, and can be written as

$$\text{KL}(P_0 \| P_t(\mathbf{h},\mathbf{J})) = tK(\mathbf{J},\mathbf{h}) + o(t),$$

where

$$K(\mathbf{J},\mathbf{h}) = \sum_{b=1}^{M} \sum_{i=1}^{N} \sum_{a=1}^{q_i} \exp\left( \frac{1}{2} \left( \left( 2y_{b,(i-1)N+a} - 1 \right) \sum_{j=1}^{N} \sum_{c=1}^{q_j} y_{b,(j-1)N+c} J_{ij}(a,c) - h_i(a) \right) \right)$$

and $y_{b,n}$ stands for the $(b, n)$ entry of matrix **Y**. Then the estimation of the parameters can be obtained by minimizing $K(\mathbf{J},\mathbf{h})$ plus $L_1$ and $L_2$ regularization factors, which can be written as

$$\left( \mathbf{J}^{\text{MPF}}, \mathbf{h}^{\text{MPF}} \right) = \begin{array}{c} \arg\min \\ \mathbf{h},\mathbf{J} \end{array} \left( K(\mathbf{J},\mathbf{h}) + \lambda_1 \sum_{i=1}^{N} \sum_{a=1}^{q_i} \sum_{j=i+1}^{N} \sum_{b=1}^{q_j} \left| J_{ij}(a,b) \right| + \lambda_2 \sum_{i=1}^{N} \sum_{a=1}^{q_i} \sum_{j=i+1}^{N} \sum_{b=1}^{q_j} J_{ij}(a,b)^2 \right), \quad (6)$$

where $\lambda_1$ and $\lambda_2$ are the coefficients of the $L_1$ and $L_2$ regularization factors respectively and are chosen manually based on the third step.

3. The third step is to choose a set of couplings and fields that satisfy Eq. (6) to initialize a gradient descent algorithm where the gradient is approximated using Markov chain Monte Carlo (MCMC) simulations. The gradient descent employs a modified RPROP algorithm[88] for each parameter set. This particular step is referred to as the Boltzmann machine-learning (BML) method, which refines the parameters obtained in the previous MPF step to achieve a more accurate model fit. During this BML process, the couplings that were forced to zero due to $L_1$ regularization in Eq. (6) remain fixed at zero in each iteration. Eventually, we adopt the parameter set described in ref. 89, such that

$$\epsilon_1 = \frac{1}{W} \sum_{i=1}^{W} \sum_{a=1}^{q_i} \frac{\left( f_i^{\text{model}}(a; \lambda_1, \lambda_2) - f_i(a, \phi^*) \right)^2}{\frac{1}{W} f_i(a, \phi^*)(1 - f_i(a, \phi^*))} \approx 1 \quad (7)$$

$$\epsilon_2 = \frac{1}{\sum_{k=1}^{W} q_k \sum_{l=k+1}^{W} q_l} \sum_{i=1}^{W} \sum_{a=1}^{q_i} \sum_{j=i+1}^{W} \sum_{b=1}^{q_j} \frac{\left( f_{ij}^{\text{model}}(a, b; \lambda_1, \lambda_2) - f_{ij}(a, b, \phi^*) \right)^2}{\frac{1}{W} f_{ij}(a, b, \phi^*)(1 - f_{ij}(a, b, \phi^*))} \approx 1, \quad (8)$$

where $f_i^{\text{model}}(a; \lambda_1, \lambda_2)$ and $f_{ij}^{\text{model}}(a, b; \lambda_1, \lambda_2)$ are the single and double mutant probabilities obtained from the model, while $f_i(a, \phi^*)$ and $f_{ij}(a, b, \phi^*)$ are the single and double mutant probabilities of the MSA after grouping with combining factor $\phi^*$. $\lambda_1$ and $\lambda_2$ are chosen to balance between overfitting and underfitting in the single and double mutant probabilities.

The MPF-BML software requires an input comprising the MSA and the patient weight of each sequence in the MSA. For model inference, all parameters were set to default values in MPF-BML software except for $L_1$ and $L_2$ regularization parameters that were set to $\lambda_1 = 10^{-4}$ and $\lambda_2 = 150$, respectively. The inferred model accurately reproduced the

statistics of the NS3 MSA (Supplementary Figure 8). These included statistics used to train the model (i.e., single mutant probabilities and double mutant probabilities) as well as the statistics predicted by the model (e.g., connected correlations and distribution of the number of mutants per sequences).

## Fitness verification

Ex-vivo experimental infectivity measurements were compiled from the literature[9,10,21,28,29] to check if our inferred NS3 prevalence landscape model is capable of capturing the underlying protein fitness landscape. We used our model to compute energies of the NS3 sequences (Eq. (1)) and compared them with their corresponding reported infectivities. Since the energy of a sequence is inversely related to its prevalence, a strong negative correlation between model-based energy and infectivity would indicate that the inferred prevalence landscape model is a good proxy of the intrinsic NS3 fitness landscape. The details of the specific fitness measurements (listed in Supplementary Data 1) from each study are presented in Supplementary Table 7. As experiments were conducted under different lab settings, we considered the weighted average of Spearman correlation coefficients from different experiments. This can be written as

$$\bar{r} = \frac{\sum_{i=1}^{q_{\exp}} Q_i r_i}{\sum_{i=1}^{q_{\exp}} Q_i},$$

where $r_i$ is the Spearman correlation coefficient between model predictions (energies) and infectivity measurements reported in experiment $i$, $Q_i$ is the number of measurements for experiment $i$, and $q_{\exp}$ is the total number of experiments.

## Conservation-only model

To compare our model with a model that ignores all interactions between residues, we defined a conservation-based maximum entropy model that is parametrized only by the "fields" **h** as follows

$$h_i(a) = \ln \frac{1 - f_i(a)}{f_i(a)}, \quad i = 1, 2, \ldots, N. \tag{9}$$

Here $f_i(a)$ is the frequency of observing amino acid $a$ at residue $i$.

## Acquisition of drug-resistant mutations for NS3-specific drugs

A total of 21 residues with DRMs from nine NS3-specific drugs used for treating HCV genotype 1a infections (listed in Table 1) were obtained from the GLUE database (http://hcv.glue.cvr.ac.uk)[5,6], as well as from other relevant studies[4,7]. An NS3 DRM is defined by an amino acid substitution at a specific residue of NS3 that is able to adversely impact the activity of a DAA in-vitro and/or in-vivo in treated patients.

## Identification of sectors using robust co-evolutionary analysis

We employed the robust co-evolutionary analysis (RoCA) method to identify 'sectors' or co-evolving groups of residues in the NS3 protein ref. 37 RoCA achieves this by performing an eigenvector-based spectral analysis on the MSA correlation matrix, followed by a data-driven random-matrix-based clustering procedure. We used the GUI-based implementation of the RoCA method, RocaSec[90], to predict NS3 sectors. Note that we chose not to use the well-known Statistical Coupling Analysis (SCA) method[91] due to its limited ability to resolve co-evolutionary structures in viral proteins, as has been demonstrated in our previous study[37].

## Visualization of interactions between top-coupled pairs of mutations

For visualizing the interactions between top-coupled pairs of mutations, we used the Circos plot. Each NS3 residue was evenly distributed along the outer space of the circle in Fig. 2. Residue numbering was

started from 1 at the 3 o'clock position and increased in the counter-clockwise direction. Only residues involving DRMs were labeled. Each link within the circle represents a pair of top-coupled mutations (ranked by the values of -$J$ from Eq. (1)). Links involving at least one DRM were shown in orange, while those between non-DRMs were shaded in gray.

## Visualization of protein crystal structures

All NS3 protein crystal structures (PDB ID: [4B6E], [3M5L], [3SU3], [3SV6], [3SUD]) were obtained from the Protein Databank (https://www.rcsb.org). The PyMOL software (https://www.pymol.org) was used for computing the distance between atoms in each protein structure and for drawing the structural figures.

## Prediction of compensatory mutations associated with SC-DRMs in different sequence backgrounds

To predict compensatory mutations connected to SC-DRMs in various sequence backgrounds (as opposed to only H77; the sequence background considered in Fig. 3), we introduced each SC-DRM into all MSA sequences lacking that mutation and computed the inferred energy change upon introducing all associated strongly coupled mutations in each selected sequence. We repeated this process for all SC-DRMs. Mutations that compensated for an SC-DRM in at least 10% of the selected sequences are listed in Supplementary Table 3.

## Statistical significance testing

We calculated the statistical significance of the number of SC-DRM residues (identified by our model) associated with a specific drug using a $p$-value. For each drug and a given number of top-coupled pairs of mutations inferred by our model (e.g., 10, 100, or 300), the p-value represents the probability that, given $j$ total DRMs associated with a specific drug, we would identify at least $i$ of them as SC-DRMs purely by chance. Let $n$ represents the number of residues involved in the top-coupled pairs, which is a subset of the $N$ total residues in the NS3 protein. In our case, $N = 515$, with 116 fully conserved residues removed. Note that in this calculation same residues involved in multiple pairs of mutations were only counted once. This $p$-value is computed as:

$$p = \sum_{q=i}^{\min(j,n)} \frac{\binom{j}{q}\binom{N-j}{n-q}}{\binom{N}{n}}. \tag{10}$$

The above equation sums up the probabilities of observing $i$ or more SC-DRMs associated with a drug using our model. If $p < 0.05$, we reject the null hypothesis that the SC-DRMs associated with a drug were observed by a random chance.

## Evolutionary simulation

We considered a Wright-Fisher-like viral evolutionary model[47] to quantify the relative ease of escape from selective pressure targeting each residue involved in the DRMs known for HCV NS3-targeting DAAs[4–7] (listed in Table 1). Similar evolutionary models have been shown to be representative of the relative ease of escape from selective pressure of immune system for HCV E2 genotype 1a and 1b[12,13], and are informative of protein structures[92–94]. As in refs. 12,13, we adopted the "escape time" metric to represent the number of generations required for mutations at a residue under selective pressure to reach a frequency of >0.5 in a fixed-size virus population.

The model set-up can be summarized as follows. The fixed virus population size was set to $M_e = 2000$ in accordance with the estimated HCV effective population size in in-host evolution[95]. For a given NS3-DRM-associated residue $i$, we started the simulation with a homogeneous population comprising copies of a randomly selected sequence from the MSA having the consensus amino acid (i.e., the

most frequent amino acid) at residue *i*. For each generation of the virus population, sequences undergo the following three steps.

1. *Mutation*. Each nucleotide in the sequences is randomly mutated to another nucleotide with a fixed probability $\mu = 10^{-4}$ in accordance with the HCV mutation rate reported in refs. 96,97.

2. *Selection*. Each sequence in the viral population survives with a probability calculated based on its fitness predicted from the inferred landscape (see ref. 12 for details). In addition, fitness of all sequences having the consensus amino acid at residue *i* is decreased by a fixed value *b*. This models the selective pressure exerted by a drug at residue *i* and provides a selective advantage to the sequences having a mutation at this residue. *b* was set as the largest value of the field parameter **h** in the inferred fitness landscape.

3. *Random sampling*. A standard multinomial sampling process, parameterized by the survival probabilities calculated in the previous step and $M_e$, is performed to generate the next generation of the virus population. The above three steps (mutation, selection, and random sampling) are repeated until the frequency of sequences having a mutation at residue *i* exceeds 0.5 in the population and the corresponding number of generations is recorded. This number is considered the time (generation) it took for the virus to escape from selective pressure at residue *i*. We re-ran this procedure multiple times (100) using the same initial sequence, as well as for multiple distinct initial sequences (25), yielding a total of 2500 values). The mean of the number of generations recorded over all these simulation runs represented the escape time associated with residue *i* (listed in Supplementary Data 3).

### Acquisition of efficacy data for NS3-specific drugs

A total of 22 studies reporting efficacy data of nine NS3-specific drugs for treating patients infected with HCV genotype 1 were included (listed in Supplementary Table 6[48–69]). In each study, efficacy of a drug was reported as the proportion of patients with SVR for 12 or 24 weeks after the end of the treatment. We used the weighted average of SVR rates associated with a drug to represent its aggregated efficacy.

### Reporting summary

Further information on research design is available in the Nature Portfolio Reporting Summary linked to this article.

## Data availability

For inferring the maximum entropy-based fitness landscape model, NS3 subtype 1a aligned sequences (coverage ≥99%) were downloaded from the publicly available HCV GLUE sequence database, http://hcv.glue.cvr.ac.uk. For validation of the inferred NS3 1a fitness landscape model, the ex-vivo experimental fitness (infectivity) measurements were compiled from five literature reports[9,10,21,28,29]. Information of drug resistant mutations of nine NS3-specific drugs used for treating HCV genotype 1a infections were obtained from the HCV GLUE database http://hcv.glue.cvr.ac.uk, as well as from two relevant literature studies[4–7]. All NS3 1a protein crystal structures (PDB ID: [4B6E], [3M5L], [3SU3], [3SV6], [3SUD]) used in the analysis were obtained from the Protein Databank (https://www.rcsb.org). For the drug efficacy analysis, efficacy data of nine NS3-specific drugs used for treating patients infected with HCV genotype 1a was compiled from 22 literature studies[48–69]. All data used in this work has been provided in the supplementary data files and is publicly available as of the date of publication. The infectivity measurements for NS3, used for correlating with predictions from the fitness landscape model, are included in Supplementary Data 1. Accession numbers of NS3 sequences used for inferring the model are listed in Supplementary Data 2. The mean escape time predicted by the in-host evolutionary model for each residue with DRMs is provided in Supplementary Data 3. Source data

for all figures are provided with this paper. Source data are provided with this paper.

## Code availability

Scripts for reproducing the results are available at https://github.com/hangzhangust/HCV_NS3[98]. The GUI-based software implementation of the MPF-BML method[33], used for inferring the fitness landscape parameters, is available at https://github.com/ahmedaq/MPF-BML-GUI[86]. The GUI-implementation of the robust co-evolutionary analysis approach, RocaSec, is available at https://github.com/ahmedaq/RocaSec[90]. For computing the distance between atoms in each protein structure and for drawing the structural figures, the PyMOL software (https://www.pymol.org) was used. All statistical analyses in this work were performed using MATLAB R2021a. Any additional information related to the data reported in this paper is available from the lead contact upon request.

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

## Acknowledgements

This work was supported by the Hong Kong Research Grants Council (grant numbers 16204121 [A.A.Q.] and 16204519 [A.A.Q.]), and Australian

Research Council Future Fellowship funded by the Australian Government (project number FT200100928 [M.R.M.]).

## Author contributions

A.A.Q. and M.R.M. conceptualized the idea and designed the research. H.Z. performed the research and generated the figures and tables. All authors analyzed the data and wrote the manuscript.

## Competing interests

The authors declare no competing interests.
