## [Peer Review File · Nature Communications]

REVIEWER COMMENTS

Reviewer #1 (Remarks to the Author):

The work of Zhang et al. aims to understand the evolutionary reasons connected with drug resistance of Direct-acting Antiviral Agents (DAA) in HCV, the virus responsible for Hepatitis C which leads to liver disease and cancer. The authors utilize a maximum-entropy model of sequence prevalence using sequences of the NS3 protein, which is a target of DAA, as a proxy for fitness and to evaluate the effect of mutations as well as to identify the strongest couplings that could be involved in Drug Resistance Mutations (DRMs). Sequences are obtained from the GLUE database where information about patients and in some cases sustained virological responses (SVR) are available. With this model Zheng et al. are able to correlate the probability of the global model to experimental fitness as well as associating top coupled residues with DRMs. Such pairs seem to have enrichment with several aspects of resistance, for example they involve residues that are known to be DRMs with higher significance than random pairs. Also, they notice some specific pairs and residues for which there exist experimental evidence of their involvement in resistance. Zhang et al. also compare SC-DRMs with residues that bind drugs in NS3 and conclude that again such residues tend to be enriched with SC-DRMs. In order to test if the model based on epistatic interactions helps to explain resistance, they include this fitness proxy into a Wright-Fisher like model, where they reveal “escape times” for mutations connected to DRM. They show that those residues that are involved in stronger epistatic interactions are able to escape faster given the selection pressure of the drugs. This provides further evidence that the role of epistasis is important in the development of DRMs, since in many cases looking directly at the mutations introduce deleterious effects to NS3.

This interesting study, showcases how global sequence models, in the style of Direct Coupling Analysis, can be useful to understand viral fitness as it has been shown for other systems like HIV as well as for several other protein bacterial systems. Although intuitive, the introduction of epistatic effects in a quantitative way has been a powerful tool to understand the complexity of evolution. My assessment is that this study provides new support for the validity of such models, and specifically shows its potential for biomedical modeling. I find that the main contribution of this study is the potential to apply this quantitative approach to forecast potential DRMs, although this work is only a first step towards that. My assessment is that results and correlations are well supported but somehow lack specificity, they focus on statistical trends and do not show results that might help us identify more mechanistic aspects of these epistatic effects. I also felt that the methods lack details as most of the core developments were done in previous studies from other groups or the authors. This affects the conceptual innovation of the results by making it look like a derivative follow up study. Having said that, I feel that the authors could revise their manuscript to improve its specificity and to integrate more analysis and methodological details to make this study more self-contained. I have a series of questions of suggestions that if clarified could help improve in those aspects.

General Comments and Questions

1. The authors use the term validation and demonstration throughout the manuscript. I felt that in some cases by “validation” they actually mean “correlation” and by “demonstration” they actually mean “support”. For example, when the maximum entropy model is compared against 45 infectivity measurements, showing a negative correlation, this is presented as a validation when in fact is a correlation analysis. Or when the evolutionary model is created, the authors state that this demonstrates how the DRMs are involved in compensatory interactions, when in fact this is a modeling scheme that provides support for this notion rather than a demonstration.

2. The authors propose an interesting case when they discuss mutations that are disruptive to NS3 but in fact lead to resistance and then propose that the fitness cost might be compensated by other mutations. I think the authors should improve specificity of this hypothesis and actually check all those mutations and search for direct evidence on those compensatory changes in the sequences they are analyzing. A few examples showing this could enhance the general nature of the results in this study, in favor of a more direct analysis on the actual changes that lead to resistance.

3. There is a statement in the first line of page 3 that says that residues involving SC-DRMs are in contact in the NS3 protein structure. This result is a known result in the field shown in the development of Direct Coupling Analysis (DCA), where directly coupled residues tend to be in contact in 3D structure for many protein families. This should probably be mentioned and referenced in the paper.

4. The authors discuss a coevolutionary approach that identifies “sectors”. Is this related to Statistical Coupling Analysis (SCA) developed in Ranganathan lab? Given that the authors don’t provide much detail in the methods section, they should clarify if these sectors are a different methodology than the highly cited work by Ranganathan otherwise also cite Ranganathan’s papers.

5. Figure 4b-4c are hard to interpret, the difference between “DRMs” and “all-DRMs” is not clear, please re-state this to make it clearer to the readers.

6. In my opinion, the most interesting contribution of this work, is the potential use of these models to predict mutations that might confer resistance. I think the authors should spend more time describing this possibility and provide more details on how this could be achieved.

7. In sequence data processing, 2167 sequences were excluded because they were not associated to patients. Since this is a big fraction of the total number, I wonder if they could still be relevant for parameter inference even if the patient weight is set to 1 or any other fraction. An analysis on the effect

on correlation to fitness by including these sequences would be interesting. Also, what do the authors mean by outlying sequences?

8. In model construction, when mentioning that this model and its energy has been used in the past, I think that the work of the Onuchic group, Weigt group, Marks and Morcos groups should be acknowledged for the development of DCA and later the Weigt and Morcos groups for the development of evolutionary models that account for epistasis.

9. Fig. S1 includes connected correlations and number of mutants per sequences. It was not clear to me how these two concepts are defined. Please clarify in the methods section.

10. For the conservation model, why not use $h_i(a)$ that is inferred from the model in Eq. 1? Is there any noticeable difference compared to the definition based on frequency counts?

11. In the definition of the NS3 DRM, the authors mention GLUE, is the definition of the DRM the same as the one of GLUE? or did the authors define it themselves? If so, please provide more details.

12. For the calculations of statistical significance, the authors use an $N=631$, but their global model uses only $N = 515$ after exclusion of conserved residues. Could the authors justify the use of $N=631$ instead of $N=515$? This difference could have an effect on the calculation of significance.

13. In the evolutionary simulation, the authors mention a “consensus amino acid” but do not define it. Please provide a proper definition of this. Is it the more popular residue?

14. In the evolutionary model, the algorithm provides a selective advantage to the sequences having a mutation at residue i , but does this mutation the actual known DRM or it could be any mutation?

15. The simulations were run 100 times for the same initial sequence, and for multiple initial sequences (25). Just to clarify, for each of those 25 initial sequences there were a 100 steps of the evolutionary process?

Minor details

1. In section C (Enrichments of SC-DRMs in NS3 drugs) when stating that non-SC-DRMs are not significantly enriched in drugs, please add (3/9) to make it consistent with the previous statement
2. Page 5, line 144. The statement is vague, please provide more details or rephrase. The same goes for the last sentence in line 159.
3. Figure 5b is not readable in black and white, please make sure to modify it to correct this.
4. Page 4, line 115. Replace “domians” with “domains”

Reviewer #3 (Remarks to the Author):

The manuscript by Zhang et al. reports on an interesting investigation on the role of epistasis in resistance against direct-acting antivirals targeting the HCV NS3 protein. The work is based on a pairwise model (Potts model) well known to reproduce accurately the prevalence landscape of protein sequences and on the use of a Wright-Fisher like evolutionary model. The insight provided by these calculations is highly significant, as it allows to give a dynamical characterization to drug resistant mutations in terms of escape time. Overall, the work sheds light on the relevance of epistasis in the emergence of drug resistant mutations and, to some extent, it clarifies the mechanism of escape from pressure. The manuscript is well written and describes the approach in sufficient details to ensure reproducibility of the results. Given the significance of the results, which are of potential interest to a large community of scientists, I recommend publication of this manuscript in the present form.

Reviewer #4 (Remarks to the Author):

This is a very interesting and well-written article using a previously developed computational approach to identify epistatic interactions between HCV drug resistance mutations (DRMs) and then to identify groups of Strongly coupled DRMs (SC-DRMs), and estimate their impact on the speed of drug resistance evolution. The work is conceptually interesting, original, potentially relevant, and well described.

I have however some concerns that should be addressed:

1) To assess the clinical relevance of this work it would be important if the authors could argue in more detail to what extent DRM to DAA represents a substantial problem from a clinical or public health perspective.

2) If I understand correctly the in-silico model (lines 285) used to derive the epistatic interactions (J_{ij}), this is strongly based on the assumption that the population from which the sequences were sampled is well-mixed and in an equilibrium state. This is particularly problematic for DRMs for several reasons:

a) The recent (positive) selective pressure for DRM, making the equilibrium assumption implausible.

b) The fluctuating selection for DRM, with some patients (receiving a specific DAA) providing an environment selecting for DRMs and others (e.g. untreated patients) representing an environment selecting against those DRMs. This could also lead to artefactual estimates of epistatic effects if for example DRMs M_1 , M_2 , M_3 etc are selected for in treated patients, but are selected against in (the vast majority) of untreated patients, then this will lead to a statistical association (linkage disequilibrium) between these mutations even if they have no epistatic interaction (because treatment acts as a confounder, e.g. $M_1 \leftarrow T \rightarrow M_2$, where T is presence of treatment, M_i the presence of mutation i). As the model does not take into account the effect of confounding by treatment it will misinterpret this association as the result of an epistatic interaction and hence estimate $J_{12} > 0$ even if there is no real/biological interaction.

c) In addition there might be a confounding by subpopulation/geography: Different subpopulations of patients will have different probabilities of receiving a given DAA, but will also differ in terms of the mutation frequencies (simply because of "phylogeographic" reasons): for example if population A is more likely to receive a given DAA (selecting for mutation M_1) than population B, but also has a higher frequency of a mutation M_2 (simply because of phylogeography), then M_1 and M_2 will again be correlated, which will be misinterpreted as an epistatic interaction.

It would be important to provide a thorough justification (ideally with additional sensitivity analyses and adjustments) that the presented results are not merely an artefact of these types of confounding.

3) Given the above concerns, it is somewhat reassuring that the authors could validate the in-silico derived fitness landscape see Figure 3 by comparing predicted with experimentally measured fitness. Here it was however not clear to me, to what extent these fitness effects were due to DRMs (where the assumptions underlining the in-silico approach seem particularly problematic, as outlined above) or whether these involve mostly other mutations (where these assumptions are less problematic). Ideally only the first type of data (i.e. sequences that differ only by presence/absence of DRM) would be used. It would also be good if the authors could specify whether the model with epistatic effects provides a significantly better correlation than the model ignoring epistatic effects (the authors mention that the correlation is stronger $r = -0.79$ vs $r = -0.55$; but it is unclear what the uncertainty/CI/variation of these estimates is and whether this difference is significant [I guess it is]; perhaps consider a bootstrapping approach?)

4)The p-value computation (line 334-) was not fully clear to me. please clarify.

RESPONSE TO REVIEWERS' COMMENTS

Epistatic interactions promote resistance against direct-acting antivirals targeting the HCV NS3 protein (NCOMMS-23-02381)

Reviewer #1 (Remarks to the Author):

The work of Zhang et al. aims to understand the evolutionary reasons connected with drug resistance of Direct-acting Antiviral Agents (DAA) in HCV, the virus responsible for Hepatitis C which leads to liver disease and cancer. The authors utilize a maximum-entropy model of sequence prevalence using sequences of the NS3 protein, which is a target of DAA, as a proxy for fitness and to evaluate the effect of mutations as well as to identify the strongest couplings that could be involved in Drug Resistance Mutations (DRMs). Sequences are obtained from the GLUE database where information about patients and in some cases sustained virological responses (SVR) are available. With this model Zhang et al. are able to correlate the probability of the global model to experimental fitness as well as associating top coupled residues with DRMs. Such pairs seem to have enrichment with several aspects of resistance, for example they involve residues that are known to be DRMs with higher significance than random pairs. Also, they notice some specific pairs and residues for which there exist experimental evidence of their involvement in resistance. Zhang et al. also compare SC-DRMs with residues that bind drugs in NS3 and conclude that again such residues tend to be enriched with SC-DRMs. In order to test if the model based on epistatic interactions helps to explain resistance, they include this fitness proxy into a Wright-Fisher like model, where they reveal "escape times" for mutations connected to DRM. They show that those residues that are involved in stronger epistatic interactions are able to escape faster given the selection pressure of the drugs. This provides further evidence that the role of epistasis is important in the development of DRMs, since in many cases looking directly at the mutations introduce deleterious effects to NS3.

This interesting study, showcases how global sequence models, in the style of Direct Coupling Analysis, can be useful to understand viral fitness as it has been shown for other systems like HIV as well as for several other protein bacterial systems. Although intuitive, the introduction of epistatic effects in a quantitative way has been a powerful tool to understand the complexity of evolution. My assessment is that this study provides new support for the validity of such models, and specifically shows its potential for biomedical modeling. I find that the main contribution of this study is the potential to apply this quantitative approach to forecast potential DRMs, although this work is only a first step towards that. My assessment is that results and correlations are well supported but somehow lack specificity, they focus on statistical trends and do not show results that might help us identify more mechanistic aspects of these epistatic effects. I also felt that the methods lack details as most of the core developments were done in previous studies from other groups or the authors. This affects the conceptual innovation of the results by making it look like a derivative follow up study. Having said that, I feel that the authors could revise their manuscript to improve its specificity and to integrate more analysis and methodological details to make this study more self-contained. I have a series of questions of suggestions that if clarified could help improve in those aspects.

Response:

Thank you for the thorough and positive assessment of our work. In light of the constructive feedback, we have revised the manuscript to address the concerns raised.

First, we now delve deeper into specific cases of epistatic interactions, highlight their importance, and draw connections with existing knowledge. This additional analysis should aid the reader in better understanding the implications and potential applications of our work. We have now also provided a more detailed explanation of our methodology in the revised manuscript (highlighted in blue). While we acknowledge that much of the core method development has been established in previous studies, we have made an effort to ensure that our study is self-contained and accessible to readers who may not be familiar with the prior work.

Overall, we believe that the revisions we have made to the manuscript have significantly improved the specificity, methodological details, and overall quality of our study.

Our point-by-point responses are detailed below.

General Comments and Questions

1. *The authors use the term validation and demonstration throughout the manuscript. I felt that in some cases by “validation” they actually mean “correlation” and by “demonstration” they actually mean “support”. For example, when the maximum entropy model is compared against 45 infectivity measurements, showing a negative correlation, this is presented as a validation when in fact is a correlation analysis. Or when the evolutionary model is created, the authors state that this demonstrates how the DRMs are involved in compensatory interactions, when in fact this is a modeling scheme that provides support for this notion rather than a demonstration.*

Response:

We have now replaced the terms “validation” and “demonstration” with “correlation” and “support”, respectively.

The text updated in response to this comment is highlighted in blue in the revised manuscript at Lines 20-23, 81, 238-239, 305, and 334-335, and in the Fig 1. caption.

2. *The authors propose an interesting case when they discuss mutations that are disruptive to NS3 but in fact lead to resistance and then propose that the fitness cost might be compensated by other mutations. I think the authors should improve specificity of this hypothesis and actually check all those mutations and search for direct evidence on those compensatory changes in the sequences they are analyzing. A few examples showing this could enhance the general nature of the results in this study, in favor of a more direct analysis on the actual changes that lead to resistance.*

Thank you for the comment. We have performed further analysis to test whether our landscape can capture specific compensatory effects associated with strongly coupled drug resistant mutations (SC-DRMs) that we selected based on our model.

Experimental data derived from in vivo or in vitro studies offers the most direct evidence for compensatory mutations associated with NS3 DRMs. However, such data is currently limited. In vivo evidence is available for the SC-DRM Q80K which has been reported to co-occur with the A91S mutation among individuals who experience HCV treatment failure ². This compensatory interaction has also been observed in vitro in the H77 strain ². Experimental evidence of SC-DRM D168E being compensated by Q41R ¹ has also been reported for the H77 strain. These two pairs of compensatory mutations were both associated with large values of $-J_{ij}$; Q80K and A91S was ranked 60th, and D168E and Q41R was ranked 1st (Fig. 3a).

We further investigated the mutational interactions predicted by our model for these two SC-DRMs, Q80K and D168E. We specifically examined the energy changes in the H77 strain bearing the D168E or Q80K mutants (denoted H77_{D168E} and H77_{Q80K} respectively) upon introducing all possible mutations. A negative energy change indicates increased fitness, whereas positive change suggests a fitness reduction. Strikingly, our model predicted that the Q41R and A91S mutations yielded the second-most negative energy change compared to all other mutations in the H77_{D168E} and H77_{Q80K} strains, respectively (Fig. R1). This outcome is consistent with the documented compensatory roles of these mutations for DRMs D168E and Q80K, and points to the specificity of our model in describing epistatic compensatory pathways.

Encouraged by these findings, we extended our analysis to predict specific compensatory mutations associated with other SC-DRMs. We explored compensatory mutations connected to SC-DRMs in various sequence backgrounds (as opposed to only H77; the sequence background considered in the analysis above). We introduced each SC-DRM into all MSA sequences lacking that mutation and computed the inferred energy change upon introducing all associated strongly coupled mutations in each selected sequence. We repeated this process for all SC-DRMs. In Table R1, we present mutations that compensated for an SC-DRM in at least 10% of the selected sequences. Interestingly, 168E and 41R were found to be compensatory (for each other) for all selected sequence backgrounds, while 91S compensated for 80K in approximately 23% of sequence backgrounds. Furthermore, we identified potential compensatory mutations for SC-DRMs 36L, 55A, 122C/G, and 177V. These identify specific

targets for future experimental studies.

Fig. R1: Histogram of the change in energy observed by all single mutations X in the H77 strain carrying the (a) D168E mutant and (b) Q80K mutant. Energy($H77_{D168E}$) and Energy($H77_{Q80K}$) are the predicted energy for the H77 strain carrying the D168E and Q80K mutant. The predicted energy for the H77 strain carrying the D168E mutant and an additional single mutation X, as well as for the H77 strain carrying the Q80K mutant and an additional single mutation X, are given by Energy($H77_{D168E+X}$) and Energy($H77_{Q80K+X}$), respectively.

Table R1: **List of top-coupled mutations that are predicted to be compensatory for SC-DRMs.** Each row shows the SC-DRM, the number of MSA sequences lacking the SC-DRM, the associated compensatory mutation (among top 300 pairs of mutations with large values of $-J_{ij}$), and the percentage of sequences where the associated mutation was found to compensate for the SC-DRM.

SC-DRM	Total number of selected sequence backgrounds	Associated compensatory mutation	Percentage of sequence backgrounds where mutation has a compensatory effect
41R	7362	168E	100%
168E	7318	41R	100%
170V	7011	174S	73.2%
80K	4553	615V	61.1%
122G	6953	174S	60.2%
122C	7207	174S	51.8%
55A	7250	40T	29.6%
80K	4553	91S	22.7%
36L	7292	197Y	11.5%
122C	7207	197Y	10.3%
122C	6953	318T	10%

We have included a new section in Results entitled, “Model predictions correlate with known NS3 DRM compensation data,” and a new section in Methods entitled, “Prediction of compensatory mutations associated with SC-DRMs in different sequence backgrounds,” which incorporate the results and methodology described above.

3. *There is a statement in the first line of page 3 that says that residues involving SC-DRMs are in contact in the NS3 protein structure. This result is a known result in the field shown in the development of Direct Coupling Analysis (DCA), where directly coupled residues tend to be in contact in 3D structure for many protein families. This should probably be mentioned and referenced in the paper.*

Response:

DCA (ref. ³) is indeed a powerful method that has been extensively used to predict residue-residue contacts in proteins. We now refer to it appropriately in the revised manuscript.

The modified text is in Lines 106-107 in the revised manuscript.

4. *The authors discuss a coevolutionary approach that identifies “sectors”. Is this related to Statistical Coupling Analysis (SCA) developed in Ranganathan lab? Given that the authors don’t provide much detail in the methods section, they should clarify if these sectors are a different methodology that the highly cited work by Ranganathan otherwise also cite Ranganathan’s papers.*

Response:

Our approach, named “robust co-evolutionary analysis (RoCA)³⁹”, is related to – but distinct from – the Statistical Coupling Analysis (SCA) method developed by the Ranganathan lab⁴. Both RoCA and SCA are based on the notion of eigenvector-based spectral analysis, however they operate on different correlation matrices, while RoCA additionally applies a data-driven random-matrix-based clustering procedure. In our previous work ³, we gave a direct comparison of RoCA and SCA, and found that while SCA is successful in identifying co-evolutionary structure for certain protein families, it does not resolve the co-evolutionary structure in HIV and HCV proteins. In contrast, the RoCA method, which applies a highly robust random-matrix based algorithm to learn co-evolutionary structure, can identify sectors that are shown to distinctly associate with often unique functional or structural domains for HIV and HCV viral proteins ^{3,5,6}. Based on these prior results, we had only referred to the RoCA method in the manuscript. We now clarify differences between the RoCA method and SCA in the Methods section of the revised manuscript (Lines 461-468), and also appropriately cite the work of Ranganathan and co-authors.

5. *Figure 4b-4c are hard to interpret, the difference between “DRMs” and “all-DRMs” is not clear, please restate this to make it clearer to the readers.*

Response:

We have now changed the color scheme to make the difference between “DRMs” and “all-DRMs” (Fig. 5; reproduced below as Fig. R3 for convenience) more obvious and we reiterate the difference in the Fig. 5 caption to make it clearer to readers.

Fig R3: Statistical significance of the number of (a) drug-specific DRMs/SC-DRMs and (b) all DRMs/SC-

DRMs in binding residues of each of the four considered drugs. Here, drug-specific DRMs/SC-DRMs are listed in Fig. 2a for each of the four drugs while all DRMs/SC-DRMs refer to the DRMs/SC-DRMs known for all drugs. The p-value measures the probability of observing by a random chance at least the observed number of DRMs or SC-DRMs among all binding residues for each drug. Results with p-value < 0.05 are marked with a star on the top of each bar.

6. *In my opinion, the most interesting contribution of this work, is the potential use of these models to predict mutations that might confer resistance. I think the authors should spend more time describing this possibility and provide more details on how this could be achieved.*

Response:

The use of the proposed model to predict drug resistant mutations is indeed interesting. To investigate this aspect, we first considered the binding residues of drugs with structures available (listed in Table R1). Of these residues (25 in total), 14 were not associated with any known DRMs. We found that mutations at four of these 14 residues (residues 78, 79, 123 and 159) were associated with strong compensatory interactions based on our model (top 300 pairs of mutations with large negative values of J_{ij}), and that at least two of these four residues were present in the binding residues of each of the four drugs, suggesting that mutations at these residues may potentially confer resistance to the respective drugs.

Table R2: **List of binding residues of drugs with known structures**. Residues that are not associated with any DRMs are shown in **bold**. Of these, residues that are associated with strong compensatory interactions based on our model (top 300 pairs of mutations with large values of J_{ij}) are also underlined.

Drug	Binding residues
Danoprevir	41, 42 , 43, 55, 57 , 58 , 78 , 79 , 80, 81 , 123 , 132, 135 , 136 , 137 , 138, 139 , 154 , 155, 156, 157 , 158, 159 , 168
Vaniprevir	41, 42 , 43, 55, 57 , 58 , 78 , 79 , 80, 81 , 123 , 132, 135 , 136 , 137 , 138, 139 , 154 , 155, 156, 157 , 158, 159 , 168
Telaprevir	41, 42 , 43, 55, 57 , 81 , 123 , 132, 135 , 136 , 137 , 138, 139 , 154 , 155, 156, 157 , 158, 159 , 168
Grazoprevir	41, 42 , 43, 55, 56, 57 , 58 , 78 , 81 , 123 , 132, 135 , 136 , 137 , 138, 139 , 154 , 155, 156, 157 , 158, 159 , 168

We have added a paragraph related to this point in the Results section (Lines 188-195) and Table R2 has been added as Supplementary Table S5.

7. *In sequence data processing, 2167 sequences were excluded because they were not associated to patients. Since this is a big fraction of the total number, I wonder if they could still be relevant for parameter inference even if the patient weight is set to 1 or any other fraction. An analysis on the effect on correlation to fitness by including these sequences would be interesting. Also, what do the authors mean by outlying sequences?*

Response:

This is a valid query. We removed sequences with no patient information to reduce patient bias in the model. Including these sequences with a patient weight of 1 reduces the correlation between sequence energies of the inferred model and in-vitro fitness measurements from $r = -0.79$ for the original model (Fig. 1) to $r = -0.56$ (Fig. R4), highlighting the significance of sequence reweighting using patient information. This suggests that it is better to discard sequences with no patient information. Additionally, we observed that a model (based on the 7370 sequences considered in our original manuscript) that ignores sequence reweighting completely (i.e., assumes each sequence is sampled from a different patient) also provides a weaker correlation with in-vitro fitness measurements ($r = -0.41$, Fig. R5). This further emphasizes the importance of the sequence reweighting step in eliminating patient bias.

Fig. R4: Correlation between the sequence energy obtained from newly inferred model and in-vitro infectivity measurements.

Fig. R5: Correlation between the sequence energy obtained from inferred model using sequences with weight of each sequence set to 1 and in-vitro infectivity measurements.

We appreciate the confusion related to outlying sequences. In brief, we first constructed a pair-wise similarity matrix of the sequences, where the (i,j) th entry of the matrix represents the fraction of amino acids that are the same across sequence i and j . Next, we conducted principal component analysis (PCA) based on the similarity matrix and investigated the first and second principal components (PCs). We considered all those sequences as outliers which appeared at more than 3 scaled median absolute deviations away from the median of either the first or second PC ⁷. The scaled median absolute deviation is given by:

$$c * \text{median} (\text{abs} (A_i - \text{median} (A))),$$

where A is the first/second PC, and A_i is the i -th element in the first/second PC,

$c = -1/(\sqrt{2} \times \text{erfcinv} (3/2)) \approx 1.482$, and $\text{erfcinv}()$ is the inverse complementary error function.

We now clarify this point in Methods (Lines 360-365).

8. *In model construction, when mentioning that this model and its energy has been used in the past, I think that the work of the Onuchic group, Weigt group, Marks and Morcos groups should be acknowledged for the development of DCA and later the Weigt and Morcos groups for the development of evolutionary models that account for epistasis.*

Response:

Thank you for this feedback regarding acknowledgment of previous works in this field⁹⁻¹². As suggested, we now acknowledge contributions of groups that developed the DCA method, as well as those that developed residue-residue interaction based evolutionary models. We cite the relevant publications⁸⁻¹¹ in the revised manuscript (Lines 106-107).

9. *Fig. S1 includes connected correlations and number of mutants per sequences. It was not clear to me how these two concepts are defined. Please clarify in the methods section.*

Response:

Connected correlations represent correlations which cannot be explained by lower order mutant probabilities. It is given by $f_{ij}(a, b) - f_i(a)f_j(b)$, where $f_i(a)$ is the probability of observing mutant a at residue i while f_{ij} is the probability of simultaneously observing mutants a and b at residues i and j respectively. The number of mutants per sequence is the number of amino acids that are different in that sequence from those of the consensus sequence (sequence constructed with the most-frequent amino acid at each residue).

We now clarify these two terms in the caption of Supplementary Fig. S8.

10. *For the conservation model, why not use $h_i(a)$ that is inferred from the model in Eq. 1? Is there any noticeable difference compared to the definition based on frequency counts?*

Response:

The fields (h_s) and couplings (J_s) are inferred jointly in our maximum entropy model (see Eq. (2) in the paper). Thus, the inferred fields cannot be used to directly construct a conservation-only model.

To determine the effect of incorporating interactions in the model, it is essential to compare a maximum entropy model that ignores pairwise interactions with one that considers them. When pairwise interactions are ignored, the maximum entropy model solution only requires inferring fields and can be obtained using a closed-form solution (Eq. 9 in the manuscript; reproduced here for easy reference):

$$h_i(a) = \ln \frac{1 - f_i(a)}{f_i(a)}, \quad i = 1, 2, \dots, N.$$

We refer to this model as the conservation-only model. These details are included in Methods.

11. *In the definition of the NS3 DRM, the authors mention GLUE, is the definition of the DRM the same as the one of GLUE? or did the authors define it themselves? If so, please provide more details.*

Response:

We have adopted the same definition of DRM as used by GLUE and other papers from which we curated DRMs¹²⁻¹⁵. Specifically, we define a DRM as an amino acid substitution that is capable of negatively affecting the activity of DAAs either in vitro or in vivo in treated patients.

We now clarify this in the Introduction section (Lines 31-33).

12. *For the calculations of statistical significance, the authors use an N=631, but their global model uses only N = 515 after exclusion of conserved residues. Could the authors justify the use of N=631 instead of N=515? This difference could have an effect on the calculation of significance.*

Response:

Thanks for raising this point. We used N=631 instead of N=515 because we wished to include the residue 156 in our analysis. Residue 156 is associated with every drug that we considered in this study, but it was fully conserved based on the available sequence data. Nevertheless, if we exclude conserved residues from our analysis and use N=515 to compute statistical significance, all results remain largely consistent. For the SC-DRMs (Fig. 2b), the association reaches statistical significance for 5/9 drugs ($p < 0.05$) and for one additional drug with marginal statistical significance ($p = 0.05$), as compared to previous results in which 6/9 drugs were associated with $p < 0.05$. As for the non-SC-DRMs, the association reaches statistical significance for 2/9 drugs in comparison to 3/9 drugs previously. Taken together, these results are consistent with our observation that SC-DRMs are statistically significantly enriched in most drugs, while non-SC-DRMs are generally not. Moreover, six binding residues of drugs with known structures (residues 57, 137, 139, 156, 157 and 158; Table R1) were fully conserved. Excluding these residues from statistical significance calculation (Fig. 2) also did not alter the qualitative results.

We now exclude fully conserved residues from all analysis and have used N=515 in calculating statistical significance of all results. All figures (Figs. 2, 5, 6, S3 and S4) have been revised in the manuscript accordingly.

13. *In the evolutionary simulation, the authors mention a “consensus amino acid” but do not define it. Please provide a proper definition of this. Is it the more popular residue?*

Response:

Thanks for pointing this out. A consensus amino acid refers to the most prevalent amino acid at a residue, i.e., the one with the highest frequency.

The modified part is in Lines 511-512 and is shown in blue.

14. *In the evolutionary model, the algorithm provides a selective advantage to the sequences having a mutation at residue i , but does this mutation the actual known DRM or it could be any mutation?*

Response:

In our simulations, we gave a selective advantage to sequences that had any mutation at a residue associated with a DRM. This was done because multiple drug-resistant mutations are often associated with a single residue, such as 80K/G/H/L/R, 122C/G/N, and 155D/G/I. This suggests that mutations that are different from the known DRMs at a residue may also potentially lead to the development of drug resistance.

Importantly, we used the same simulation setting for computing the time required to escape the drug-induced selective pressure by SC-DRMs and other DRMs (Fig. 6). By keeping the conditions for simulating the escape dynamics of these two sets of DRMs the same, we were able to compare them fairly without exhaustively simulating the time required for specific mutations to occur. This is because observing such mutations in our simulations could take a very long time.

15. The simulations were run 100 times for the same initial sequence, and for multiple initial sequences (25). Just to clarify, for each of those 25 initial sequences there were a 100 steps of the evolutionary process?

Response:

To clarify, we conducted 100 Monte Carlo runs for each of the 25 initial sequences. In each run, the sequence population underwent several rounds of mutation, selection, and random sampling steps until the mutations at the considered residue i became dominant (frequency > 0.5) in the population.

We have now clarified this in Line 529.

Minor details

1. In section C (Enrichments of SC-DRMs in NS3 drugs) when stating that non-SC-DRMs are not significantly enriched in drugs, please add (3/9) to make it consistent with the previous statement

Response:

As suggested, we have now added 1/9 (after excluding all conserved residues) to make it consistent.

2. Page 5, line 144. The statement is vague, please provide more details or rephrase. The same goes for the last sentence in line 159.

Response:

We have now revised the text (Lines 157-160, 186-187) to make it clearer to readers.

3. Figure 5b is not readable in black and white, please make sure to modify it to correct this.

Response: As suggested, we have now changed the color scheme to make it easier to read in black and white, reproduced below for convenience.

Fig. R6: Escape time of residues involved in NS3 DRMs. **(a)** Comparison between escape time of residues involved in SC-DRMs and the remaining residues involved in DRMs. In each box plot, the middle line indicates the median, the edges of the box represent the first and third quartiles, and whiskers extend to span a 1.5 interquartile range from the edges. The reported p-value was calculated using the two-sided Mann-Whitney test. **(b)** Individual escape time of residues involved in DRMs of the NS3 protein. SC-DRMs are shown in blue and the remaining DRMs in orange.

4. Page 4, line 115. Replace “domians” with “domains”

Response:

This typo has been corrected. We have conducted a thorough review of the paper to ensure that there are no other such errors.

Reviewer #3 (Remarks to the Author):

The manuscript by Zhang et al. reports on an interesting investigation on the role of epistasis in resistance against direct-acting antivirals targeting the HCV NS3 protein. The work is based on an pairwise model (Potts model) well known to reproduce accurately the prevalence landscape of protein sequences and on the use of a Wright-Fisher like evolutionary model. The insight provided by these calculations is highly significant, as it allows to give a dynamical characterization to drug resistant mutations in terms of escape time. Overall, the work sheds light on the relevance of epistasis in the emergence of drug resistant mutations and, to some extent, it clarifies the mechanism of escape from pressure. The manuscript is well written and describes the approach in sufficient details to ensure reproducibility of the results. Given the significance of the results, which are of potential interest to a large community of scientists, I recommend publication of this manuscript in the present form.

Response:

Thank you for your positive and encouraging feedback on our manuscript. We appreciate your recommendation.

Reviewer #4 (Remarks to the Author):

This is a very interesting and well-written article using a previously developed computational approach to identify epistatic interactions between HCV drug resistance mutations (DRMs) and then to identify groups of Strongly coupled DRMs (SC-DRMs) and estimate their impact on the speed of drug resistance evolution. The work is conceptually interesting, original, potentially relevant, and well described.

I have however some concerns that should be addressed:

- 1. To assess the clinical relevance of this work it would be important if the authors could argue in more detail to what extent DRM to DAA represents a substantial problem from a clinical or public health perspective.*

Response:

Thank you for the comment.

DRMs are important in the treatment of chronic HCV patients with DAAs for both clinical and public health reasons. Firstly, DAAs are highly effective against HCV, but DRMs can cause treatment failure as the virus can become resistant to the drugs being used. Patients who have been previously treated with DAAs and have not achieved sustained virologic response (SVR) are at increased risk of developing DRMs, which can limit the effectiveness of subsequent treatment¹⁶. Secondly, studies show that a significant percentage of patients with chronic HCV infection who have failed previous DAA treatment have DRMs, with prevalence rates ranging from 20–90% depending on the specific DAA used and the type of DRMs^{17–19}. This further supports the notion that DRMs may accumulate in the viral population and further impede virological cure. Thirdly, the accumulation of DRMs in drug-experienced patients with chronic HCV infection can be problematic from a clinical and public health perspective. This is because these patients may have limited treatment options available to them, as the development of DRMs can render some DAAs ineffective²⁰. This can lead to prolonged infection, an increased risk of liver cirrhosis, and ultimately liver failure. In addition, DRMs can be transmitted to other individuals, potentially leading to the spread of drug-resistant strains of HCV²¹.

Additionally, HIV, like HCV, can develop DRMs that reduce the effectiveness of antiretroviral therapy (ART) which is used to manage HIV infection. According to the WHO, 75% of HIV-infected patients were receiving ART at the end of 2021²², compared to less than 20% of HCV-infected patients receiving DAAs²³. A study found that the proportion of people with HIV DRMs increased from 11% in 2001 to 29% in 2016, with those who had previously taken ART being more likely to have DRMs²⁴. With more widespread usage of DAAs for HCV, HCV DRMs may become similarly clinically relevant as HIV DRMs in the future.

We have emphasized these key points in the Discussion (Lines 226-236) to provide more clarity on the clinical relevance of DRMs to DAAs in the context of HCV treatment.

- 2. If I understand correctly the in-silico model (lines 285) used to derive the epistatic interactions (J_{ij}), this is strongly based on the assumption that the population from which the sequences were sampled is well-mixed and in an equilibrium state. This is particularly problematic for DRMs for several reasons:*

a) The recent (positive) selective pressure for DRM, making the equilibrium assumption implausible.

b) The fluctuating selection for DRM, with some patients (receiving a specific DAA) providing an environment selecting for DRMs and others (e.g. untreated patients) representing an environment selecting against those DRMs. This could also lead to artefactual estimates of epistatic effects if for example DRMs M_1 , M_2 , M_3 etc are selected for in treated patients, but are selected against in (the vast majority) of untreated patients, then this will lead to a statistical association (linkage disequilibrium) between these mutations even if they have no epistatic interaction (because treatment acts as a confounder, e.g. $M_1 \leftarrow T \rightarrow M_2$, where T is presence of treatment, M_i the presence of mutation i). As the model does not take into account the effect of confounding by treatment it will misinterpret this association as the result of an epistatic interaction and hence estimate $J_{12} > 0$ even if there is no

real/biological interaction.

c) In addition there might be a confounding by subpopulation/geography: Different subpopulations of patients will have different probabilities of receiving a given DAA, but will also differ in terms of the mutation frequencies (simply because of "phylogeographic" reasons): for example if population A is more likely to receive a given DAA (selecting for mutation M1) than population B, but also has a higher frequency of a mutation M2 (simply because of phylogeography), then M1 and M2 will again be correlated, which will be misinterpreted as an epistatic interaction.

It would be important to provide a thorough justification (ideally with additional sensitivity analyses and adjustments) that the presented results are not merely an artefact of these types of confounding.

Response:

Thanks for the insightful comments. We agree that various factors, such as the selective pressure from host immune responses and the recent use of direct-acting antivirals (DAAs), could potentially confound the inference of a reliable fitness landscape model from population-level sequence data. Nevertheless, these factors appear to minimally impact our model inference.

A main reason why potential DAA-induced selective pressures are not anticipated to strongly bias our results is that DAAs are currently only available to a limited fraction of HCV-infected individuals (less than 20%; ref.³⁶). To examine this more explicitly for the specific NS3 data set that we used (comprising 7370 sequences), we investigated the 58 papers from which these sequences were reported. This analysis revealed that the large majority of the sequences (5877 sequences) were indeed from drug-naïve patients. We quantified the potential bias of drug-induced pressure by comparing statistical properties of the complete dataset (7370 sequences) used to infer our model with those of the drug-naïve subset (5877 sequences). This analysis revealed a strong correlation ($r > 0.9$, Fig. R1) between the mutation frequencies and pairwise mutation frequencies, used to infer models, in both datasets. We also constructed a maximum entropy model using only the drug-naïve sequences and found that the predicted sequence energies from the drug-naïve model exhibited a correlation ($r = -0.70$, Fig. R2) with the in-vitro fitness measurements which was comparable to the correlation observed with the complete dataset. This analysis, overall, suggests potential data biases due to drug-induced selection pressures to be weak.

We agree with the reviewer that, with prevalent DAA administration, phylogeographic effects and differences in drug distribution patterns could potentially present confounders that may be misinterpreted as epistasis. This is not a significant factor in our analysis, however. Most importantly, the large majority of the data is drug-naïve, as indicated above. Moreover, specific epistatic interactions predicted by our model (associated with large values of $-J_{ij}$) have been observed in vivo as well as in vitro^{1,2}, which provides supports to the ability of our model to predict epistasis (please refer to Section II C of the revised manuscript for further details. This section has been added in response to Comment 2 from Reviewer 1). Furthermore, since we focus on subtype 1a, most of our data (6802 out of 7370 sequences) was sequenced from the US, where DAA distribution patterns are anticipated to be similar across regions, and therefore, to not present strong geographical biases affecting our data set.

The strong correlation observed between the inferred prevalence landscape and in vitro fitness measurements of HCV NS3 (Fig. 1) substantiates that our inferred landscape is not strongly influenced of confounding factors and serves as a reasonably accurate representation of the underlying intrinsic fitness landscape of NS3. While such a simple relationship between prevalence and fitness has been reported for HCV proteins E2 and NS5B²⁵⁻²⁷, as well as several HIV proteins²⁸⁻³⁰, it has not been observed for the surface proteins of the influenza A virus³¹ or the polio virus³². A mechanistic rationale for this relationship has been previously proposed for HIV proteins, with three key factors identified³³: (i) a diverse and largely ineffective immune response due to host genetic diversity, (ii) reversion to the ancestral (fitter) sequence upon transmission to a new host, and (iii) the absence of robust and effective natural or vaccine-induced herd memory responses, which would shift the virus away from the steady state. Although HCV differs from HIV, it shares several similarities and may also involve these factors. Particularly, most sequences are sampled from chronic patients (with acute HCV infections being predominantly asymptomatic) and with NS3 being a target of T cells³⁴, it is likely that NS3 experiences diverse and ineffective immune responses in such patients. Reversion to the consensus amino acid upon

HCV transmission to a new host has also been documented³⁵.

Fig. R1: Correlation of single mutant probabilities (left panel) and double mutant probabilities (right panel) between sequences from all patients (7370 sequences) and the subset of drug-naïve patients (5877 sequences).

Fig. R2: Correlation between model predicted energies and experimental fitness measurements compiled from different studies (mentioned in the legend). The model was inferred using sequences from the subset of drug-naïve patients (5877 sequences). Normalization of both fitness measurements and predicted model energies was performed by subtracting the mean from each data set and dividing by its standard deviation.

In the revised manuscript, we have added two paragraphs in Discussion (Lines 247-271) explaining the rationale behind obtaining a meaningful fitness landscape of HCV NS3 from population-level sequence data despite the presence of potentially confounding selective factors.

3. Given the above concerns, it is somewhat reassuring that the authors could validate the in-silico derived fitness landscape see Figure 3 by comparing predicted with experimentally measured fitness. Here it was however not clear to me, to what extent these fitness effects were due to DRMs (where the assumptions underlining the in-silico approach seem particularly problematic, as outlined above) or whether these involve mostly other mutations (where these assumptions are less problematic). Ideally only the first type of data (i.e. sequences that differ only by presence/absence of DRM) would be used. It would also be good if the authors could specify whether the model with epistatic effects provides a significantly better correlation than the model ignoring epistatic effects (the authors mention that the correlation is stronger $r = -0.79$ vs $r = -0.55$; but it is unclear what the uncertainty/CI/variation of these

estimates is and whether this difference is significant [I guess it is]; perhaps consider a bootstrapping approach?)

Response:

Of the 45 fitness measurements we compiled from different experimental studies, 36 were associated with DRMs. The strong correlation between our model's predictions and experimental fitness measurements indeed is reassuring that even though our model is trained largely on drug-naïve sequences (as indicated in the response above), it can still accurately capture the intrinsic effect of DRMs. This is because while DRMs are enriched in patients that fail DAA treatment^{37,38}, they are not exclusively observed in such individuals and have been observed in drug-naïve patients also. We additionally checked the correlation between our model predictions and these 36 fitness measurements that are exclusively associated with DRMs. We found this correlation to be strong as well ($r = -0.72$, Fig. R3), further corroborating the capability of our inferred model to capture the fitness of the virus.

Fig. R3: Correlation between the model predicted energies and 36 experimental fitness measurements that are associated with DRMs. These fitness measurements were compiled from different studies that are mentioned in the legend. Normalization of both fitness measurements and predicted model energies was performed by subtracting the mean from each data set and dividing by its standard deviation.

In the revised manuscript, we have added a paragraph in Discussion (mentioned in blue at Lines 272-279) explaining the above result.

In regard to the statistical variation of the correlation estimates, we have now applied a bootstrapping approach, as suggested, to investigate if our inferred maximum entropy model that considers epistatic effects provides a statistically significantly stronger correlation than a model that does not consider these effects (conservation-only model). Specifically, we inferred maximum-entropy and conservation-only models by randomly drawing sequences with replacement from the original data. As shown in Fig. R4, maximum entropy models provided statistically significantly stronger correlation between the model predicted energies and experimental fitness values than conservation-only models ($p = 1.70 \times 10^{-4}$, two-sided Mann-Whitney test; number of bootstrapping samples = 10).

We have added Fig. R4 in the revised manuscript as Supplementary Fig. 1 and the related text is mentioned in Results at Lines 79-82.

Fig. R4: Robustness of the correlation observed between model energies and experimental fitness values. Results are shown for the maximum-entropy model that considers epistatic interaction and for the conservation-only model that ignores epistasis. The sequence data used for inferring each model was generated by a standard bootstrap approach for ten samples.

4. The *p*-value computation (line 334-) was not fully clear to me. please clarify.

Response:

The *p*-value represents the probability that, given *j* total DRMs associated with a specific drug, we would identify at least *i* of them as SC-DRMs purely by chance. SC-DRMs are found in the top-coupled pairs of mutations (e.g., top 300) in our model. Here, *n* represents the total number of residues involved in the top-coupled pairs, which is a subset of the *N* total residues in the NS3 protein. In our case, *N* = 515, with 116 fully conserved residues removed. Note that in this calculation same residues involved in multiple pairs of mutations were only counted once. This *p*-value is computed as:

$$p = \sum_{q=i}^{\min(j,n)} \frac{\binom{j}{q} \binom{N-j}{n-q}}{\binom{N}{n}}.$$

The above equation sums up the probabilities of observing *i* or more SC-DRMs associated with a drug using our model. If *p* < 0.05, we rejected the null hypothesis that the SC-DRMs associated with a drug were observed by a random chance.

We would like to note that in response to reviewer 1's comment we have changed *N* = 631 to *N* = 515 to include only the non-conserved residues in all analyses.

We now explain this point further in the revised manuscript. The related text has been added in Methods at Lines 491-499.

References

1. Dultz, G. *et al.* Extended interaction networks with HCV protease NS3-4A substrates explain the lack of adaptive capability against protease inhibitors. *Journal of Biological Chemistry* **295**, 13862–13874 (2020).
2. Dultz, G. *et al.* Epistatic interactions promote persistence of NS3-Q80K in HCV infection by compensating for protein folding instability. *Journal of Biological Chemistry* **297**, 101031 (2021).
3. Quadeer Ahmed Abdul AND Morales-Jimenez, D. A. N. D. M. M. R. Co-evolution networks of HIV/HCV are modular with direct association to structure and function. *PLoS Comput Biol* **14**, 1–29 (2018).
4. Halabi, N., Rivoire, O., Leibler, S. & Ranganathan, R. Protein Sectors: Evolutionary Units of Three-Dimensional Structure. *Cell* **138**, 774–786 (2009).
5. Dahirel, V. *et al.* Coordinate linkage of HIV evolution reveals regions of immunological vulnerability. *Proceedings of the National Academy of Sciences* **108**, 11530–11535 (2011).
6. Quadeer, A. A. *et al.* Statistical Linkage Analysis of Substitutions in Patient-Derived Sequences of Genotype 1a Hepatitis C Virus Nonstructural Protein 3 Exposes Targets for Immunogen Design. *J Virol* **88**, 7628–7644 (2014).
7. Leys, C., Ley, C., Klein, O., Bernard, P. & Licata, L. Detecting outliers: Do not use standard deviation around the mean, use absolute deviation around the median. *J Exp Soc Psychol* **49**, 764–766 (2013).
8. Hopf, T. A. *et al.* Sequence co-evolution gives 3D contacts and structures of protein complexes. *Elife* **3**, (2014).
9. Morcos, F., Jana, B., Hwa, T. & Onuchic, J. N. Coevolutionary signals across protein lineages help capture multiple protein conformations. *Proceedings of the National Academy of Sciences* **110**, 20533–20538 (2013).
10. Marks, D. S. *et al.* Protein 3D structure computed from evolutionary sequence variation. *PLoS One* **6**, (2011).
11. Weigt, M., White, R. A., Szurmant, H., Hoch, J. A. & Hwa, T. Identification of direct residue contacts in protein-protein interaction by message passing. *Proceedings of the National Academy of Sciences* **106**, 67–72 (2008).
12. Singer Joshua B. and Thomson, E. C. and M. J. and H. J. and G. R. J. GLUE: A flexible software system for virus sequence data. *BMC Bioinformatics* **19**, 532 (2018).
13. Singer, J. *et al.* Interpreting Viral Deep Sequencing Data with GLUE. *Viruses* **11**, 323 (2019).
14. Sorbo, M. C. *et al.* Hepatitis C virus drug resistance associated substitutions and their clinical relevance: Update 2018. *Drug Resistance Updates* **37**, 17–39 (2018).
15. Romano, K. P., Ali, A., Royer, W. E. & Schiffer, C. A. Drug resistance against HCV NS3/4A inhibitors is defined by the balance of substrate recognition versus inhibitor binding. *Proceedings of the National Academy of Sciences* **107**, 20986–20991 (2010).
16. Pawlotsky, J. M. Hepatitis C Virus Resistance to Direct-Acting Antiviral Drugs in Interferon-Free Regimens. *Gastroenterology* vol. 151 70–86 Preprint at <https://doi.org/10.1053/j.gastro.2016.04.003> (2016).
17. Izumi, N. *et al.* Sofosbuvir–velpatasvir plus ribavirin in Japanese patients with genotype 1 or 2 hepatitis C who failed direct-acting antivirals. *Hepato Int* **12**, 356–367 (2018).
18. Mawatari, S. *et al.* The co-existence of NS5A and NS5B resistance-associated substitutions is associated with virologic failure in Hepatitis C Virus genotype 1 patients treated with sofosbuvir and ledipasvir. *PLoS One* **13**, (2018).
19. Sarrazin, C. *et al.* Prevalence of Resistance-Associated Substitutions in HCV NS5A, NS5B, or NS3 and Outcomes of Treatment With Ledipasvir and Sofosbuvir. *Gastroenterology* **151**, 501-512.e1 (2016).
20. Spengler, U. Direct antiviral agents (DAAs) - A new age in the treatment of hepatitis C virus infection. *Pharmacology and Therapeutics* vol. 183 118–126 Preprint at <https://doi.org/10.1016/j.pharmthera.2017.10.009> (2018).
21. Deng, H. *et al.* Dynamic changes of HCV genomes under selective pressure from DAAs therapy in relapsed patients. *Virus Res* **302**, (2021).
22. World Health Organization. HIV and AIDS-Fact Sheet. <https://www.who.int/en/news-room/fact-sheets/detail/hiv-aids> (2021).

23. World Health Organization. Hepatitis C-Fact Sheet. <https://www.who.int/news-room/fact-sheets/detail/hepatitis-c> (2021).
24. Gregson, J. *et al.* Global epidemiology of drug resistance after failure of WHO recommended first-line regimens for adult HIV-1 infection: A multicentre retrospective cohort study. *Lancet Infect Dis* **16**, 565–575 (2016).
25. Zhang, H., Quadeer, A. A. & McKay, M. R. Evolutionary modeling reveals enhanced mutational flexibility of HCV subtype 1b compared with 1a. *iScience* **25**, (2022).
26. Quadeer, A. A., Louie, R. H. Y. & McKay, M. R. Identifying immunologically-vulnerable regions of the HCV E2 glycoprotein and broadly neutralizing antibodies that target them. *Nat Commun* **10**, 2073 (2019).
27. Hart, G. R. & Ferguson, A. L. Empirical fitness models for hepatitis C virus immunogen design. *Phys Biol* **12**, 066006 (2015).
28. Ferguson, A. L. *et al.* Translating HIV Sequences into Quantitative Fitness Landscapes Predicts Viral Vulnerabilities for Rational Immunogen Design. *Immunity* **38**, 606–617 (2013).
29. Mann, J. K. *et al.* The Fitness Landscape of HIV-1 Gag: Advanced Modeling Approaches and Validation of Model Predictions by In Vitro Testing. *PLoS Comput Biol* **10**, e1003776 (2014).
30. Louie, R. H. Y., Kaczorowski, K. J., Barton, J. P., Chakraborty, A. K. & McKay, M. R. Fitness landscape of the human immunodeficiency virus envelope protein that is targeted by antibodies. *Proceedings of the National Academy of Sciences* **115**, E564–E573 (2018).
31. Łuksza, M. & Lässig, M. A predictive fitness model for influenza. *Nature* **507**, 57–61 (2014).
32. Quadeer, A. A., Barton, J. P., Chakraborty, A. K. & McKay, M. R. Deconvolving mutational patterns of poliovirus outbreaks reveals its intrinsic fitness landscape. *Nat Commun* **11**, 377 (2020).
33. Shekhar, K. *et al.* Spin models inferred from patient-derived viral sequence data faithfully describe HIV fitness landscapes. *Phys. Rev. E* **88**, 062705 (2013).
34. Ward, S., Lauer, G., Isba, R., Walker, B. & Klenerman, P. Cellular immune responses against hepatitis C virus: the evidence base 2002. *Clin Exp Immunol* **128**, 195–203 (2002).
35. Ansari, M. A. *et al.* Genome-to-genome analysis highlights the effect of the human innate and adaptive immune systems on the hepatitis C virus. *Nat Genet* **49**, 666–673 (2017).
36. World Health Organization. Hepatitis C, Fact sheet. (2021).
37. Paolucci, S. *et al.* Baseline and Breakthrough Resistance Mutations in HCV Patients Failing DAAs. *Sci Rep* **7**, (2017).
38. Raj, V. S. *et al.* Identification of HCV Resistant Variants against Direct Acting Antivirals in Plasma and Liver of Treatment Naïve Patients. *Sci Rep* **7**, (2017).

REVIEWERS' COMMENTS

Reviewer #1 (Remarks to the Author):

After reviewing the rebuttal letter and the new version of the manuscript I was quite satisfied with the clear and detailed responses from the authors. I was able to clarify all my questions and I feel the paper now benefits from a set of explanations and new details that will make their work much more self-contained. The changes were extensive and I was glad to see very clear and quantitative responses to my queries. I also had the chance to see the responses to other reviewers and in my opinion the responses are as clear and thorough as for my questions. Given the improved nature of the manuscript and my initial positive assessment of this work, I support the publication of this work that I believe will be of interest to several scientific communities and in general for its biomedical implications.

Reviewer #4 (Remarks to the Author):

I would like to thank the authors for thoroughly addressing my comments. I have no further concerns and think that this work will make a valuable contribution to Nature Communications